# Probabilistically Robust Recourse: Navigating the Trade-offs between Costs and Robustness in Algorithmic Recourse

**Martin Pawelczyk**[1,*] **Teresa Datta**[2], **Johannes van-den-Heuvel**[1]
**Gjergji Kasneci**[1], **Himabindu Lakkaraju**[2]
[1]University of Tübingen, Germany
[2]Harvard University, US

## Abstract

As machine learning models are increasingly being employed to make consequential decisions in real-world settings, it becomes critical to ensure that individuals who are adversely impacted (e.g., loan denied) by the predictions of these models are provided with a means for recourse. While several approaches have been proposed to construct recourses for affected individuals, the recourses output by these methods either achieve low costs (i.e., ease-of-implementation) or robustness to small perturbations (i.e., noisy implementations of recourses), but not both due to the inherent trade-offs between the recourse costs and robustness. Furthermore, prior approaches do not provide end users with any agency over navigating the aforementioned trade-offs. In this work, we address the above challenges by proposing the first algorithmic framework which enables users to effectively manage the recourse cost vs. robustness trade-offs. More specifically, our framework Probabilistically ROBust rEcourse (`PROBE`) lets users choose the probability with which a recourse could get invalidated (recourse invalidation rate) if small changes are made to the recourse i.e., the recourse is implemented somewhat noisily. To this end, we propose a novel objective function which simultaneously minimizes the gap between the achieved (resulting) and desired recourse invalidation rates, minimizes recourse costs, and also ensures that the resulting recourse achieves a positive model prediction. We develop novel theoretical results to characterize the recourse invalidation rates corresponding to any given instance w.r.t. different classes of underlying models (e.g., linear models, tree based models etc.), and leverage these results to efficiently optimize the proposed objective. Experimental evaluation with multiple real world datasets demonstrates the efficacy of the proposed framework.

## 1 Introduction

Machine learning (ML) models are increasingly being deployed to make a variety of consequential decisions in domains such as finance, healthcare, and policy. Consequently, there is a growing emphasis on designing tools and techniques which can provide *recourse* to individuals who have been adversely impacted by the predictions of these models (Voigt & Von dem Bussche, 2017). For example, when an individual is denied a loan by a model employed by a bank, they should be informed about the reasons for this decision and what can be done to reverse it. To this end, several approaches in recent literature tackled the problem of providing recourse by generating counterfactual explanations (Wachter et al., 2018; Ustun et al., 2019; Karimi et al., 2020a). which highlight what features need to be changed and by how much to flip a model's prediction. While the aforementioned approaches output low cost recourses that are easy to implement (i.e., the corresponding counterfactuals are close to the original instances), the resulting recourses suffer from a severe lack of robustness as demonstrated by prior works (Pawelczyk et al., 2020b; Rawal et al., 2021). For example, the aforementioned approaches generate recourses which do not remain

---

*Corresponding author: martin.pawelczyk@uni-tuebingen.de

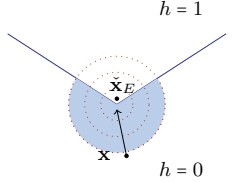
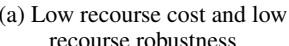

(a) Low recourse cost and low recourse robustness

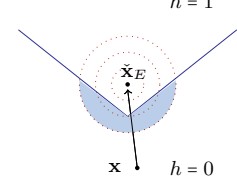
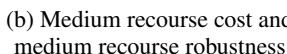

(b) Medium recourse cost and medium recourse robustness

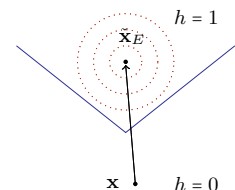

(c) High recourse cost and high recourse robustness

Figure 1: Pictorial representation of the recourses (counterfactuals) output by various state-of-the-art recourse methods and our framework. The blue line is the decision boundary, and the shaded areas correspond to the regions of recourse invalidation. Fig. 1a shows the recourse output by approaches such as Wachter et al. (2018) where both the recourse cost as well as robustness are low. Fig. 1c shows the recourse output by approaches such as Dominguez-Olmedo et al. (2022) where both the recourse cost and robustness are high. Fig. 1b shows the recourse output by our framework PROBE in response to user input requesting an intermediate level of recourse robustness.

valid (i.e., result in a positive model prediction) if/when small changes are made to them (See Figure 1a). However, recourses are often noisily implemented in real world settings as noted by prior research (Björkegren et al., 2020). For instance, an individual who was asked to increase their salary by $500 may get a promotion which comes with a raise of $505 or even $499.95.

Prior works by Upadhyay et al. (2021) and Dominguez-Olmedo et al. (2022) proposed methods to address some of the aforementioned challenges and generate robust recourses. While the former constructed recourses that are robust to small shifts in the underlying model, the latter constructed recourses that are robust to small input perturbations. These approaches adapted the classic minimax objective functions commonly employed in adversarial robustness and robust optimization literature to the setting of algorithmic recourse, and used gradient descent style approaches to optimize these functions. In an attempt to generate recourses that are robust to either small shifts in the model or to small input perturbations, the above approaches find recourses that are farther away from the underlying model's decision boundaries (Tsipras et al., 2018; Raghunathan et al., 2019), thereby increasing the recourse costs i.e., the distance between the counterfactuals (recourses) and the original instances. Higher cost recourses are harder to implement for end users as they are farther away from the original instance vectors (current user profiles). Putting it all together, the aforementioned approaches generate robust recourses that are often high in cost and are therefore harder to implement (See Figure 1c), without providing end users with any say in the matter. In practice, each individual user may have a different preference for navigating the trade-offs between recourse costs and robustness – e.g., some users may be willing to tolerate additional cost to avail more robustness to noisy responses, whereas other users may not.

In this work, we address the aforementioned challenges by proposing a novel algorithmic framework called Probabilistically ROBust rEcourse (PROBE) which enables end users to effectively manage the recourse cost vs. robustness trade-offs by letting users choose the probability with which a recourse could get invalidated (recourse invalidation rate) if small changes are made to the recourse i.e., the recourse is implemented somewhat noisily (See Figure 1b). To the best of our knowledge, this work is the first to formulate and address the problem of enabling users to navigate the trade-offs between recourse costs and robustness. Our framework can ensure that a resulting recourse is invalidated at most $r\%$ of the time when it is noisily implemented, where $r$ is provided as input by the end user requesting recourse. To operationalize this, we propose a novel objective function which simultaneously minimizes the gap between the achieved (resulting) and desired recourse invalidation rates, minimizes recourse costs, and also ensures that the resulting recourse achieves a positive model prediction. We develop novel theoretical results to characterize the recourse invalidation rates corresponding to any given instance w.r.t. different classes of underlying models (e.g., linear models, tree based models etc.), and leverage these results to efficiently optimize the proposed objective.

We also carried out extensive experimentation with multiple real-world datasets. Our empirical analysis not only validated our theoretical results, but also demonstrated the efficacy of our proposed framework. More specifically, we found that our framework PROBE generates recourses that are not

only three times less costly than the recourses output by the baseline approaches (Upadhyay et al., 2021; Dominguez-Olmedo et al., 2022), but also more robust (See Table 1). Further, our framework `PROBE` reliably identified low cost recourses at various target recourse invalidation rates $r$ in case of both linear and non-linear classifiers (See Table 1 and Figure 4). On the other hand, the baseline approaches were not only ill-suited to achieve target recourse invalidation rates but also had trouble finding recourses in case of non-linear classifiers.

## 2 RELATED WORK

**Algorithmic Approaches to Recourse.** As discussed earlier, several approaches have been proposed in literature to provide recourse to individuals who have been negatively impacted by model predictions (Tolomei et al., 2017; Laugel et al., 2017; Wachter et al., 2018; Ustun et al., 2019; Van Looveren & Klaise, 2019; Pawelczyk et al., 2020a; Mahajan et al., 2019; Mothilal et al., 2020; Karimi et al., 2020a; Rawal & Lakkaraju, 2020; Karimi et al., 2020b; Dandl et al., 2020; Antorán et al., 2021; Spooner et al., 2021). These approaches can be roughly categorized along the following dimensions (Verma et al., 2020): *type of the underlying predictive model* (e.g., tree based vs. differentiable classifier), whether they encourage *sparsity* in counterfactuals (i.e., only a small number of features should be changed), whether counterfactuals should lie on the *data manifold* and whether the underlying *causal relationships* should be accounted for when generating counterfactuals, All these approaches generate recourses assuming that the prescribed recourses will be correctly implemented by users.

**Robustness of Algorithmic Recourse.** Prior works have focused on determining the extent to which recourses remain robust to the choice of the underlying model (Pawelczyk et al., 2020b; Black et al., 2021; Pawelczyk et al., 2023), shifts or changes in the underlying models (Rawal et al., 2021; Upadhyay et al., 2021), or small perturbations to the input instances (Artelt et al., 2021; Dominguez-Olmedo et al., 2022; Slack et al., 2021). To address these problems, these works have primarily proposed adversarial inimax objectives to minimize the worst-case loss over a plausible set of instance perturbations for linear models to generate robust recourses (Upadhyay et al., 2021; Dominguez-Olmedo et al., 2022), which are known to generate overly costly recourse suggestions.

In contrast to the aforementioned approaches our work focuses on a user-driven framework for navigating the trade-offs between recourse costs and robustness to noisy responses by suggesting a novel probabilistic recourse framework. To this end, we present several algorithms that enable us to handle both linear and non-linear models (e.g., deep neural networks, tree based models) effectively, resulting in better recourse cost/invalidation rate tradeoffs compared to both Upadhyay et al. (2021) and Dominguez-Olmedo et al. (2022).

## 3 PRELIMINARIES

Here, we first discuss the generic formulation leveraged by several state-of-the-art recourse methods including Wachter et al. (2018). We then define the notion of recourse invalidation rate formally.

### 3.1 ALGORITHMIC RECOURSE: GENERAL FORMULATION

**Notation** Let $h : \mathcal{X} \to \mathcal{Y}$ denote a classifier which maps features $\mathbf{x} \in \mathcal{X} \subseteq \mathbb{R}^d$ to labels $\mathcal{Y}$. Let $\mathcal{Y} = \{0, 1\}$ where 0 and 1 denote an unfavorable outcome (e.g., loan denied) and a favorable outcome (e.g., loan approved), respectively. We also define $h(\mathbf{x})=g(f(\mathbf{x}))$, where $f : \mathcal{X} \to \mathbb{R}$ is a differentiable scoring function (e.g., logit scoring function) and $g : \mathbb{R} \to \mathcal{Y}$ an activation function that maps logit scores to binary labels. Throughout the remainder of this work we will use $g(u) = \mathbb{I}[u > \xi]$, where $\xi$ is a decision rule in logit space. W.l.o.g. we will set $\xi = 0$.

Counterfactual (CF) explanation methods provide recourses by identifying which attributes to change for reversing an unfavorable model prediction. Since counterfactuals that propose changes to features such as gender are not actionable, we restrict the search space to ensure that only actionable changes are allowed. Let $\mathcal{A}$ denote the set of actionable counterfactuals. For a given predictive model $h$, and a predefined cost function $d_c : \mathbb{R}^d \to \mathbb{R}_+$, the problem of finding a counterfactual explanation $\check{\mathbf{x}} = \mathbf{x} + \boldsymbol{\delta}$ for an instance $\mathbf{x} \in \mathbb{R}^d$ is expressed by the following optimization problem:

$$\check{\mathbf{x}} = \underset{\mathbf{x}' \in \mathcal{A}}{\arg\min} \; \ell\big(h(\mathbf{x}'), 1\big) + \lambda \cdot d_c(\mathbf{x}, \mathbf{x}'), \tag{1}$$

where $\lambda \geq 0$ is a trade-off parameter, and $\ell(\cdot, \cdot)$ is the mean-squared-error (MSE) loss.

The first term on the right-hand-side ensures that the model prediction corresponding to the counterfactual i.e., $h(\mathbf{x}')$ is close to the favorable outcome label $1$. The second term encourages low-cost recourses; for example, Wachter et al. (2018) propose $\ell_1$ or $\ell_2$ distances to ensure that the distance between the original instance $\mathbf{x}$ and the counterfactual $\check{\mathbf{x}}$ is small.

## 3.2 Defining the Recourse Invalidation Rate

In order to enable end users to effectively navigate the trade-offs between recourse costs and robustness, we let them choose the probability with which a recourse could get invalidated (recourse invalidation rate) if small changes are made to it i.e., the recourse is implemented somewhat noisily. To this end, we formally define the notion of Recourse Invalidation Rate (IR) in this section. We first introduce two key terms, namely, *prescribed recourses* and *implemented recourses*. A prescribed recourse is a recourse that was provided to an end user by some recourse method (e.g., increase salary by \$500). An implemented recourse corresponds to the recourse that the end user finally implemented (e.g., salary increment of \$505) upon being provided with the prescribed recourse. With this basic terminology in place, we now proceed to formally define the Recourse Invalidation Rate (IR) below.

**Definition 1** (Recourse Invalidation Rate). *For a given classifier $h$, the recourse invalidation rate corresponding to the counterfactual $\check{\mathbf{x}}_E = \mathbf{x} + \boldsymbol{\delta}_E$ output by a recourse method $E$ is given by:*

$$\Delta(\check{\mathbf{x}}_E) = \mathbb{E}_{\boldsymbol{\varepsilon}}\Big[\underbrace{h(\check{\mathbf{x}}_E)}_{CF\ class} - \underbrace{h(\check{\mathbf{x}}_E + \boldsymbol{\varepsilon})}_{class\ after\ response}\Big], \tag{2}$$

*where the expectation is taken with respect to a random variable $\boldsymbol{\varepsilon}$ with probability distribution $p_{\boldsymbol{\varepsilon}}$ which captures the noise in human responses.*

Since the implemented recourses do not typically match the prescribed recourses $\check{\mathbf{x}}_E$ (Björkegren et al., 2020), we add $\boldsymbol{\varepsilon}$ to model the noise in human responses. As we primarily compute recourses for individuals $\mathbf{x}$ such that $h(\mathbf{x}) = 0$, the label corresponding to the counterfactual is given by $h(\check{\mathbf{x}}_E)=1$ and therefore $\Delta \in [0, 1]$. For example, the following cases help understand our recourse invalidation rate metric better: When $\Delta=0$, then the prescribed recourse and the recourse implemented by the user agree all the time; when $\Delta=0.5$, the prescribed recourse and the implemented recourse agree half of the time, and finally, when $\Delta=1$ then the prescribed recourse and the recourse implemented by the user never agree. To illustrate our ideas, we will use our IR measure with a Gaussian probability distribution (i.e., $\boldsymbol{\varepsilon} \sim \mathcal{N}(\mathbf{0}, \sigma^2 \mathbf{I})$) to model the noise in human responses.

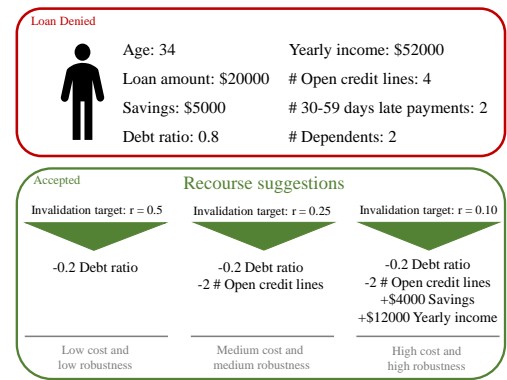

Figure 2: Practical view on navigating the cost/robustness tradeoff for a credit loan example.

## 4 Our Framework: Probabilistically Robust Recourse

Below we present our objective function, which is followed by a discussion on how to operationalize it efficiently.

## 4.1 Recourse invalidation rate aware objective

The core idea is to find a recourse $\check{\mathbf{x}}$ whose prediction at any point y within some set around $\check{\mathbf{x}}$ belongs to the positive class with probability $1 - r$. Hence, our goal is to devise an algorithm that reliably guides the recourse search towards regions of low invalidation probability while maintaining low cost recourse (see Fig. 2 for a practical example). For a fixed model, our objective reads:

$$\mathcal{L} = \lambda_1 R(\mathbf{x}'; \sigma^2 \mathbf{I}) + \lambda_2 \ell\big(f(\mathbf{x}'), s\big) + \lambda_3 d_c(\mathbf{x}', \mathbf{x}), \tag{3}$$

where $s$ is the target score for the input $\mathbf{x}$, $R(\mathbf{x}'; r, \sigma^2 \mathbf{I}) = \max(0, \Delta(\mathbf{x}'; \sigma^2 \mathbf{I}) - r)$ with $r$ being the target IR, $\Delta(\mathbf{x}'; \sigma^2 \mathbf{I})$ is the recourse invalidation rate from equation 1, $\lambda_1$ to $\lambda_3$ are the balance parameters, and $d_c$ quantifies the distance between the input and the prescribed recourse. To arrive at a output probability of $0.5$, the target score for $f(\mathbf{x})$ for a sigmoid function is $s = 0$, where the score corresponds to a $0.5$ probability for $y = 1$.

The new component $R$ is a Hinge loss encouraging that the prescribed recourse has a low probability of invalidation, and the parameter $\sigma^2$ is the uncertainty magnitude and controls the size of the neighbourhood in which the recourse has to be robust. The middle term encourages the score at the prescribed recourse $f(\check{\mathbf{x}})$ to be close to the target score $s$, while the last term promotes the distance between the input $\mathbf{x}$ and the recourse $\check{\mathbf{x}}$ to be small.

In practice, the choice of $r$ depends on the risk-aversion of the end-user. If the end-user is not confident about achieving a 'precision landing', then a rather low invalidation target should be chosen (i.e., $r < 0.5$).

## 4.2 Optimizing the recourse invalidation rate aware objective

**Algorithm 1** PROBE

**Input:** $\mathbf{x}$ s.t. $f(\mathbf{x}) < 0$, $f, \sigma^2, \lambda > 0$, $\alpha, r > 0$
**Init.:** $\mathbf{x}' = \mathbf{x}$;
Compute $\tilde{\Delta}(\mathbf{x}')$   ▷ from Theorem 1
**while** $\tilde{\Delta}(\mathbf{x}') > r$ **and** $f(\mathbf{x}') < 0$ **do**
    $\tilde{\Delta} = \texttt{ClosedFormIR}(f, \sigma^2, \mathbf{x}')$
    ▷ from Theorem 1
    $\mathbf{x}' = \mathbf{x}' - \alpha \cdot \nabla_{\mathbf{x}'} \mathcal{L}(\mathbf{x}'; \sigma^2, r, \lambda)$
        ▷ Opt. equation 3
**end while**
**Return:** $\check{\mathbf{x}} = \mathbf{x}'$

In order for the objective in equation 3 to guide us reliably towards recourses with low target invalidation rate $r$, we need to approximate the invalidation rate $\Delta(\mathbf{x}')$ at any $\mathbf{x}' \in \mathbb{R}^d$. However, such an approximation becomes non-trivial since the recourse invalidation rate, which depends on the classifier $h$, is generally non-differentiable since the classifier $h(\mathbf{x}) = I(f(\mathbf{x}) > \xi)$ as defined in Section 3 involves an indicator function acting on the score $f$. To circumvent this issue, we derive a closed-form expression for the IR using a local approximation of the predictive model $f$. The procedure suggested here remains generalizable even for non-linear models since the local behavior of a given non-linear model can often be well approximated by fitting a locally linear model (Ribeiro et al., 2016; Ustun et al., 2019).

**Theorem 1** (Closed-Form Recourse Invalidation Rate). *A first-order approximation $\tilde{\Delta}$ to the recourse invalidation rate $\Delta$ in equation 2 under Gaussian distributed noise in human responses $\varepsilon \sim \mathcal{N}(\mathbf{0}, \sigma \mathbf{I})$ is given by:*

$$\tilde{\Delta}(\check{\mathbf{x}}_E; \sigma^2 \mathbf{I}) = 1 - \Phi\left( \frac{f(\check{\mathbf{x}}_E)}{\sqrt{\nabla f(\check{\mathbf{x}}_E)^\top \sigma^2 \mathbf{I} \nabla f(\check{\mathbf{x}}_E)}} \right), \tag{4}$$

*where $\Phi$ is the CDF of the univariate standard normal distribution $\mathcal{N}(0, 1)$, $f(\check{\mathbf{x}}_E)$ denotes the logit score at $\check{\mathbf{x}}_E$ which is the recourse output by a recourse method $E$, and $h(\check{\mathbf{x}}_E) \in \{0, 1\}$.*

All theoretical proofs along with the proof to the above proposition can be found in Appendix D. In Algorithm 1, we show pseudo-code of our optimization procedure. Using gradient descent we update the recourse repeatedly until the class label flips from 0 to 1 and the IR $\tilde{\Delta}$ is smaller than the targeted invalidation rate $r$. In essence, the result in Theorem 1 serves as our regularizer since it steers recourses towards low-invalidation regions. For example, when $f(\check{\mathbf{x}}_E) = 0$, then $\tilde{\Delta} = 0.5$ since $\Phi(0) = \frac{1}{2}$. This means that the prescribed recourse and the recourse implemented by the user *agree* $50\%$ *of the time*. On the other hand, when $f(\check{\mathbf{x}}_E) \to +\infty$, then $\tilde{\Delta} \to 0$ since $\Phi \to 1$, which means that the prescribed recourse and the recourse implemented by the user *always agree*. Figure 3 demonstrates how PROBE finds recourses relative to a standard low-cost algorithm (Wachter et al., 2018).

We now leverage the recourse invalidation rate derived in Theorem 1 to show how the recourses output by Wachter et al. (2018) can be made more robust. Pawelczyk et al. (2022) provide a closed-form solution for the recourse output by Wachter et al. (2018) w.r.t. the special case of a logistic regression classifier when $d_c = \|\mathbf{x} - \mathbf{x}'\|_2$ and the MSE-loss is used. This solution takes the following form: $\check{\mathbf{x}}_{\text{Wachter}}(s) = \mathbf{x} + \frac{s - f(\mathbf{x})}{\|\nabla f(\mathbf{x})\|_2^2} \nabla f(\mathbf{x})$, where $s$ is the target logit score. More specifically, to arrive at the desired class with probability of $0.5$, the target score for a sigmoid function is $s = 0$, where the logit corresponds to a $0.5$ probability for $y = 1$. The next statement quantifies the IR of recourses output by Wachter et al. (2018).

**Proposition 1** (Exact Recourse IR). *For logistic regression, consider the recourse output by Wachter et al. (2018): $\check{\mathbf{x}}_{Wachter}(s) = \mathbf{x} + \frac{s-f(\mathbf{x})}{\|\nabla f(\mathbf{x})\|_2^2}\nabla f(\mathbf{x})$. Then the recourse invalidation rate is given by:*

$$\Delta(\check{\mathbf{x}}_{Wachter}(s); \sigma^2\mathbf{I}) = 1 - \Phi\left(\frac{s}{\sigma\|\nabla f(\mathbf{x})\|_2}\right), \tag{5}$$

*where $s$ is the target logit score.*

A recourse generated by Wachter et al. (2018) such that $f(\check{\mathbf{x}}_{\text{Wachter}}) = s = 0$ will result in $\Delta = 0.5$. To obtain recourse that is more robust to noisy responses from users, i.e., $\Delta \to 0$, the decision maker can choose a higher logit target score of $s' > s \geq 0$ since this decreases the recourse invalidation rate, i.e., $\Delta(\check{\mathbf{x}}_{\text{Wachter}}(s)) > \Delta(\check{\mathbf{x}}_{\text{Wachter}}(s'))$. The next statement makes precise how $s$ should be chosen to achieve a desired robustness level.

**Corollary 1.** *Under the conditions of Proposition 1, choosing $s_r = \sigma\|\nabla f(\mathbf{x})\|_2\Phi^{-1}(1-r)$ guarantees a recourse invalidation rate of $r$, i.e., $\Delta(\check{\mathbf{x}}_{Wachter}(s_r); \sigma^2\mathbf{I}) = r$.*

**On extensions to general noise distributions, and tree-based classifiers.** In Appendix A we present extensions of our framework to obtain (i) reliable recourses for general noise distributions and (ii) tree-based classifiers. These two cases pose non-trivial difficulties as the recourse invalidation rate is generally non-differentiable. As for the more general noise distributions, we develop a Monte-Carlo approach in appendix A.1, which relies on a differentiable approximation of the indicator function required to obtain a Monte-Carlo estimate of the invalidation rate. For tree-based classifiers, we develop a closed-form solution for the recourse invalidation rate (see Theorem 2).

### 4.3 Additional Theoretical Results

In this section, we leverage the recourse invalidation rate expression derived in the previous section to theoretically show i) that an additional cost has to be incurred to generate robust recourses in the face of noisy human responses, and ii) we derive a general upper bound on the IR which is applicable to any valid recourse provided by any method with the underlying classifier being a differentiable model.

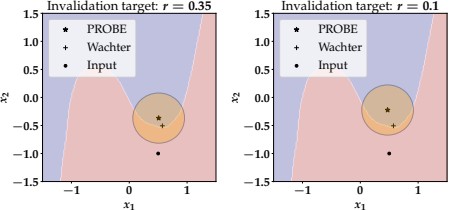

Figure 3: Navigating between high and low invalidation recourses. The circles around PROBE's recourses have radius $2\sigma$, i.e., this is the region where 95% of recourse inaccuracies fall when $\sigma^2 = 0.05$. For instance, on the left we set an invalidation target of $r = 0.35$, i.e., 35% of the recourse responses would fail under spherical inaccuracies $\boldsymbol{\varepsilon} \sim \mathcal{N}(\mathbf{0}, 0.05 \cdot \mathbf{I})$.

Next, we show that there exists a trade-off between robustness to noisy human responses and cost. To this end, we fix the target invalidation rate $r$, and ask what costs are needed to achieve a fixed level $r$:

**Proposition 2** (General Cost of Recourse). *For a linear classifier, let $r \in (0, 1)$ and let $\check{\mathbf{x}}_E = \mathbf{x} + \boldsymbol{\delta}_E$ be the output produced by some recourse method $E$ such that $h(\check{\mathbf{x}}_E) = 1$. Then the cost required to achieve a fixed invalidation target $r$ is:*

$$\|\boldsymbol{\delta}_E\|_2 = \frac{\sigma}{\omega}\big(\Phi^{-1}(1-r) - c\big), \tag{6}$$

*where $c = \frac{f(\mathbf{x})}{\sigma \cdot \|\nabla f(\mathbf{x})\|_2}$ is a constant, and $\omega > 0$ is the cosine of the angle between $\nabla f(\mathbf{x})$ and $\boldsymbol{\delta}_E$.*

From Proposition 2, we see that the target invalidation rate $r$ decreases as the recourse cost increases for a given uncertainty magnitude $\sigma^2$. To make this more precise the next statement demonstrates the cost-robustness tradeoff.

**Proposition 3** (Cost-Robustness Tradeoff). *Under the same conditions as in Proposition 2, we have $\frac{\partial \|\boldsymbol{\delta}_E\|_2}{\partial (1-r)} = \frac{\sigma}{\omega}\frac{1}{\phi(\Phi^{-1}(1-r))} > 0$, i.e., an infinitesimal increase in robustness (i.e.,$1 - r$) increases the cost of recourse by $\frac{\sigma}{\omega}\frac{1}{\phi(\Phi^{-1}(1-r))}$.*

Now, we derive a general upper bound on the recourse invalidation rate. This bound is applicable to any method $E$ that provides recourses resulting in a positive outcome.

**Proposition 4** (Upper Bound). *Let $\check{\mathbf{x}}_E$ be the output produced by some recourse method $E$ such that $h(\check{\mathbf{x}}_E) = 1$. Then, an upper bound on $\tilde{\Delta}$ from equation 4 is given by:*

$$\tilde{\Delta}(\check{\mathbf{x}}_E; \sigma^2 \mathbf{I}) \leq 1 - \Phi\left(c + \frac{\omega}{\sigma} \frac{\|\nabla f(\mathbf{x})\|_2}{\|\nabla f(\check{\mathbf{x}}_E)\|_2} \frac{\|\boldsymbol{\delta}_E\|_1}{\sqrt{\|\boldsymbol{\delta}_E\|_0}}\right),  \tag{7}$$

*where $c = \frac{f(\mathbf{x})}{\sigma \cdot \|\nabla f(\mathbf{x})\|_2}$, $\boldsymbol{\delta}_E = \check{\mathbf{x}}_E - \mathbf{x}$, and $\omega > 0$ is the cosine of the angle between $\nabla f(\mathbf{x})$ and $\boldsymbol{\delta}_E$.*

The right term in the inequality entails that the upper bound depends on the ratio of the $\ell_1$ and $\ell_0$-norms of the recourse action $\boldsymbol{\delta}_E$ provided by recourse method $E$. The higher the $\ell_1/\ell_0$ ratio of the recourse actions, the tighter the bound. The bound is tight when $\|\boldsymbol{\delta}_E\|_0$ assumes minimum value i.e., $\|\boldsymbol{\delta}_E\|_0 = 1$ since at least one feature needs to be changed to flip the model prediction.

## 5 EXPERIMENTAL EVALUATION

We now present our empirical analysis. First, we validate our theoretical results on the recourse invalidation rates across various recourse methods. Second, we study the effectiveness of PROBE at finding robust recourses in the presence of noisy human responses.

**Real-World Data and Noisy Responses.** Regarding real-world data, we use the same data sets as provided in the recourse and counterfactual explanation library CARLA (Pawelczyk et al., 2021). The *Adult* data set Dua & Graff (2017) originates from the 1994 Census database, consisting of 14 attributes and 48,842 instances. The class label indicates whether an individual has an income greater than 50,000 USD/year. The *Give Me Some Credit* (GMC) data set Kaggle-Competition (2011) is a credit scoring data set, consisting of 150,000 observations and 11 features. The class label indicates if the corresponding individual will experience financial distress within the next two years (*SeriousDlqin2yrs* is 1) or not. The *COMPAS* data set Angwin et al. (2016) contains data for more than 10,000 criminal defendants in Florida. It is used by the jurisdiction to score defendant's likelihood of re-offending. The class label indicates if the corresponding defendant is high or low risk for recidivism. All the data sets were normalized so that $\mathbf{x} \in [0, 1]^d$. Across all experiments, we add noise $\varepsilon$ to the prescribed recourse $\check{\mathbf{x}}_E$, where $\varepsilon \sim \mathcal{N}(\mathbf{0}, \sigma^2 \cdot \mathbf{I})$ and $\sigma^2 = 0.01$.

**Methods.** We compare the recourses generated by PROBE to four different baseline methods which aim to generate low-cost recourses using fundamentally different principles: AR (-LIME) uses an integer-programming-based objective Ustun et al. (2019), Wachter uses a gradient-based objective (Wachter et al., 2018), DICE uses a diversity-based objectve (Mothilal et al., 2020), and GS is based on a random search algorithm (Laugel et al., 2017). Further, we compare with methods that use adversarial minmax objectives to generate robust recourse (Dominguez-Olmedo et al., 2022; Upadhyay et al., 2021). We used the recourse implementations from CARLA (Pawelczyk et al., 2021). Following Upadhyay et al. (2021), all methods search for counterfactuals over the same set of balance parameters $\lambda \in \{0, 0.25, 0.5, 0.75, 1\}$ when applicable.

**Prediction Models.** For all data sets, we trained both ReLU-based NN models with 50 hidden layers (App. B) and a logistic regerssion (LR). All recourses were generated with respect to these classifiers.

**Measures.** We consider three measures in our evaluation: 1) We measure the *average cost* (AC) required to act upon the prescribed recourses where the average is taken with respect to all instances in the test set for which a given method provides recourse. Since all our algorithms are optimizing for the $\ell_1$-norm we use this as our cost measure. 2) We use *recourse accuracy* (RA) defined as the fraction of instances in the test set for which acting upon the prescribed recourse results in the desired prediction. 3) We compute the *average IR* across every instance in the test set. To do that, we sample 10,000 points from $\varepsilon \sim \mathcal{N}(\mathbf{0}, \sigma^2 \mathbf{I})$ for every instance and compute IR in equation 2. Then the *average IR* quantifies recourse robustness where the individual IRs are averaged over all instances from the test set for which a given method provides recourse.

### 5.1 VALIDATING OUR THEORETICAL BOUNDS

**Computing Bounds.** We empirically validate the theoretical upper bounds derived in Section 4.3. To do that, we first estimate the bounds for each instance in the test set according to Proposition 4,

| | | Adult | | | | Compas | | | | GMC | | | |
|---|---|---|---|---|---|---|---|---|---|---|---|---|---|
| | Measures | AR | Wachter | GS | PROBE | AR | Wachter | GS | PROBE | AR | Wachter | GS | PROBE |
| LR | RA (↑) | 0.98 | **1.0** | **1.0** | **1.0** | **1.0** | **1.0** | **1.0** | **1.0** | **1.0** | **1.0** | **1.0** | **1.0** |
| | AIR (↓) | $0.5 \pm 0.01$ | $0.46 \pm 0.02$ | $0.35 \pm 0.11$ | $\mathbf{0.34 \pm 0.02}$ | $0.48 \pm 0.04$ | $0.47 \pm 0.02$ | $0.3 \pm 0.18$ | $\mathbf{0.28 \pm 0.02}$ | $0.47 \pm 0.06$ | $0.45 \pm 0.03$ | $0.48 \pm 0.04$ | $\mathbf{0.24 \pm 0.01}$ |
| | AC (↓) | $\mathbf{0.55 \pm 0.4}$ | $0.62 \pm 0.43$ | $2.12 \pm 1.05$ | $1.56 \pm 0.92$ | $\mathbf{0.16 \pm 0.17}$ | $0.22 \pm 0.17$ | $0.73 \pm 0.45$ | $0.63 \pm 0.39$ | $\mathbf{0.29 \pm 0.27}$ | $0.49 \pm 0.51$ | $\mathbf{0.28 \pm 0.31}$ | $0.60 \pm 0.56$ |
| NN | RA (↑) | 0.38 | **1.0** | **1.0** | 0.99 | 0.84 | **1.0** | **1.0** | **1.0** | 0.38 | **1.0** | **1.0** | **1.0** |
| | AIR (↓) | $0.49 \pm 0.03$ | $0.5 \pm 0.02$ | $0.48 \pm 0.02$ | $\mathbf{0.35 \pm 0.01}$ | $0.34 \pm 0.09$ | $0.46 \pm 0.02$ | $0.43 \pm 0.07$ | $\mathbf{0.33 \pm 0.02}$ | $0.34 \pm 0.07$ | $0.43 \pm 0.03$ | $0.45 \pm 0.03$ | $\mathbf{0.25 \pm 0.03}$ |
| | AC (↓) | $1.05 \pm 0.22$ | $\mathbf{0.3 \pm 0.19}$ | $2.99 \pm 1.51$ | $1.43 \pm 0.49$ | $1.15 \pm 0.52$ | $\mathbf{0.2 \pm 0.16}$ | $0.81 \pm 0.45$ | $0.8 \pm 0.34$ | $0.2 \pm 0.19$ | $0.26 \pm 0.18$ | $\mathbf{0.12 \pm 0.09}$ | $0.47 \pm 0.21$ |

(a) Comparing PROBE to baseline recourse methods.

| | | Adult | | | Compas | | | GMC | | |
|---|---|---|---|---|---|---|---|---|---|---|
| | Measures | ROAR | ARAR | PROBE | ROAR | ARAR | PROBE | ROAR | ARAR | PROBE |
| LR | RA(↑) | **1.0** | **1.0** | **1.0** | **1.0** | 0.99 | **1.0** | **1.0** | **1.0** | **1.0** |
| | AIR (↓) | $\mathbf{0.0 \pm 0.0}$ | $0.02 \pm 0.01$ | $0.34 \pm 0.02$ | $\mathbf{0.0 \pm 0.0}$ | $\mathbf{0.0 \pm 0.0}$ | $0.28 \pm 0.02$ | $\mathbf{0.0 \pm 0.0}$ | $0.35 \pm 0.01$ | $0.24 \pm 0.01$ |
| | AC (↓) | $3.56 \pm 0.8$ | $2.68 \pm 0.79$ | $\mathbf{1.56 \pm 0.92}$ | $2.99 \pm 0.31$ | $1.74 \pm 0.3$ | $\mathbf{0.63 \pm 0.39}$ | $1.74 \pm 0.45$ | $1.27 \pm 0.45$ | $\mathbf{0.60 \pm 0.56}$ |
| NN | RA(↑) | 0.94 | 0.03 | **0.99** | 0.97 | 0.02 | **1.0** | 0.06 | 0.06 | **1.0** |
| | AIR (↓) | $\mathbf{0.0 \pm 0.0}$ | $0.51 \pm 0.0$ | $0.35 \pm 0.01$ | $\mathbf{0.01 \pm 0.06}$ | $0.46 \pm 0.0$ | $0.33 \pm 0.02$ | $0.3 \pm 0.21$ | $0.45 \pm 0.01$ | $\mathbf{0.25 \pm 0.03}$ |
| | AC (↓) | $19.8 \pm 3.39$ | $0.04 \pm 0.0^{*}$ | $\mathbf{1.43 \pm 0.49}$ | $6.41 \pm 1.07$ | $0.02 \pm 0.0^{*}$ | $\mathbf{0.8 \pm 0.34}$ | $0.67 \pm 0.94$ | $0.02 \pm 0.0^{*}$ | $\mathbf{0.47 \pm 0.21}$ |

(b) Comparing PROBE to adversarially robust recourse methods.

Table 1: Comparing PROBE to recourse methods from literature using recourse accuracy (RA), average recourse invalidation rate (AIR) for $\sigma^2 = 0.01$ and average cost (AC) across different recourse methods. For PROBE, we generated recourses by setting $r = 0.35$ and $\sigma^2 = 0.01$. (a): Recourses that use our framework PROBE are more robust compared to those produced by existing baselines. (b): Adversarially robust recourses are more costly than recourses output by PROBE. For ARAR and ROAR we set $\epsilon = 0.01$. *: Results with recourse accuracies less than 10% have not been considered.

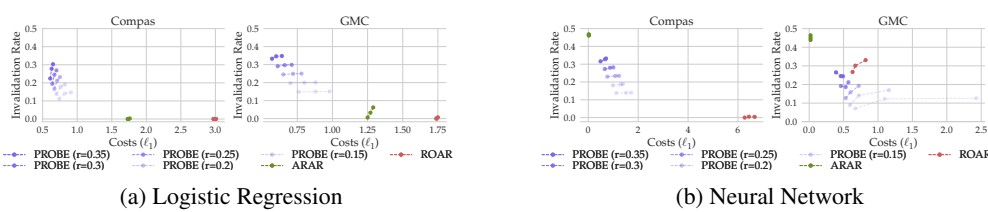

(a) Logistic Regression  (b) Neural Network

Figure 4: Comparing PROBE to adversarially robust recourse methods using pareto plots that show the tradeoff between average costs and average invalidation rate (towards bottom left indicates a better performance). For PROBE, the invalidation target is $r \in \{0.35, 0.3, 0.25, 0.20, 0.15\}$, and we generated recourses by setting $\sigma^2, \epsilon \in \{0.005, 0.01, 0.015\}$. The latter are used for ARAR and ROAR.

and compare them with the empirical estimates of the IR. The empirical IR, in turn, we obtain from Monte-Carlo estimates of the IR in equation 2; we used 10,000 samples to get a stable estimate of IR.

**Results.** In Figure 5, we validate the bounds obtained in Proposition 4 for the GMC data sets. We relegated results for the Compas and Adult data set and other values of $\sigma^2$ to Appendix C. Note that the trivial upper bound is 1 since $\Delta \leq 1$, and we see that our bounds usually lie well below this value, which suggests that our bounds are meaningful. We observe that these upper bounds are quite tight, thus providing accurate estimates of the worst case recourse invalidation rates. It is noteworthy that GS tends to provide looser bounds, since its recourses tend to have lower $\ell_1/\ell_0$ ratios; for GS, its random search procedure increases the $\ell_0$-norms of the recourse relative to the recourses output by other recourse methods. This contributes to a looser bound saying that the randomly sampled recourses by GS tend to provide looser worst-case IR estimates relative to all the other methods, which do use gradient information (e.g., Wachter , AR and PROBE).

## 5.2 EVALUATING THE PROBE FRAMEWORK

**Results.** Here, we evaluate the robustness, costs and recourse accuracy of the recourses generated by our framework PROBE relative to the baselines. We consider a recourse robust if the recourse remains valid (i.e., results in positive outcome) even after small changes are made to it (i.e., humans implement it in a noisy manner). Table 1 shows the average IR for different methods across different real world data sets and classifiers when $\sigma^2 = 0.01$. Further, in Table 1a we see that PROBE has the lowest invalidation rate across all real-world data sets and classifiers among the non-robust recourse methods, while PROBE provides the lowest cost recourses among the robust recourse methods (see

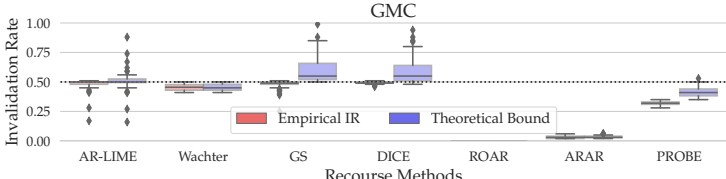

Figure 5: Verifying the theoretical upper bound from Proposition 4 on the logistic regression model. The red boxplots show the empirical recourse invalidation rates for `AR(-LIME)`, `Wachter`, `GS`, `DICE`, `ARAR` ($\epsilon = 0.01$), `ROAR` ($\epsilon = 0.01$) and `PROBE` ($r = 0.35, \sigma^2 = 0.01$). The blue boxplots show the distribution of upper bounds evaluated by plugging in the corresponding quantities (i.e., $\sigma^2$, $\omega$, etc.) into the bound. The results show no violations of our theoretical bounds. See appendix C for the full set of results.

Table 1b). We also consider if the robustness achieved by our framework is coming at an additional cost i.e., by sacrificing recourse accuracy (RA) or by increasing the average recourse cost (AC). We compute AC of the recourses output by all the algorithms and find that `PROBE` usually has the highest or second highest recourse costs, while the RA is at 100% across classifiers and data sets.

Finally, we provide a more detailed comparison between `PROBE` and the adversarially robust recourse methods `ARAR` and `ROAR`. To do so, we plot pareto frontiers in Figure 4 which demonstrate the inherent tradeoffs between the average cost of recourse and the average recourse invalidation rate computed over all recousre seeking individuals for different uncertainty magnitudes $\sigma^2, \epsilon \in \{0.005, 0.01, 0.15\}$. For `ARAR` and `ROAR` we expect to see AIRs close to 0 (by construction). However, this is only the case for the linear classifiers. Moreover, `ROAR` provide recourses with up to 3 times higher cost relative to our method `PROBE`. Note also that `ARAR` and `ROAR` have trouble finding recourses for non-linear classifiers, resulting in RA scores of around 5% in the worst case, while not being able to maintain low invalidation scores. This is likely due to the local linear approximation used by these methods. In summary, `PROBE` finds recourses for 100% of the test instances in line with the promise of having an invalidation probability of at most $r$, while being less costly than `ROAR` and `ARAR`.

**Relegated results.** The relegated experiments in Appendix C (i) demonstrate that baseline recourse methods are not robust to noisy human responses (Figures 8 - 9), (ii) verify that the targeted invalidation rates match the empirical recourse invalidation rates (Figures 13 - 15) and (iii) demonstrate the trade-off between recourse costs and robustness verifying Corollary 3 (Figures 16 - 17).

## 6 CONCLUSION

In this work, we proposed a novel algorithmic framework called Probabilistically ROBust rEcourse (PROBE) which enables end users to effectively manage the recourse cost vs. robustness trade-offs by letting users choose the probability with which a recourse could get invalidated (recourse invalidation rate) if small changes are made to the recourse i.e., the recourse is implemented somewhat noisily. To the best of our knowledge, this work is the first to formulate and address the problem of enabling users to navigate the trade-offs between recourse costs and robustness. Our framework can ensure that a resulting recourse is invalidated at most $r\%$ of the time when it is noisily implemented, where $r$ is provided as input by the end user requesting recourse. To operationalize this, we proposed a novel objective function which simultaneously minimizes the gap between the achieved (resulting) and desired recourse invalidation rates, minimizes recourse costs, and also ensures that the resulting recourse achieves a positive model prediction. We developed novel theoretical results to characterize the recourse invalidation rates corresponding to any given instance w.r.t. different classes of underlying models (e.g., linear models, tree based models etc.), and leveraged these results to efficiently optimize the proposed objective. Experimental evaluation with multiple real world datasets not only demonstrated the efficacy of the proposed framework, but also validated our theoretical findings. Our work also paves the way for several interesting future research directions in the field of algorithmic recourse. For instance, it would be interesting to build on this work to develop approaches which can generate recourses that are simultaneously robust to noisy human responses, noise in the inputs, as well as shifts in the underlying models.

ACKNOWLEDGEMENTS

We would like to thank the anonymous reviewers for their insightful feedback. This work is supported in part by the NSF awards #IIS-2008461 and #IIS-2040989, and research awards from Google, JP Morgan, Amazon, Bayer, Harvard Data Science Initiative, and D^3 Institute at Harvard. HL would like to thank Sujatha and Mohan Lakkaraju for their continued support and encouragement. The views expressed here are those of the authors and do not reflect the official policy or position of the funding agencies.

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

# A EXTENSIONS TO OTHER NOISE DISTRIBUTIONS AND TREE BASED CLASSIFIERS

## A.1 EXTENSIONS TO GENERAL NOISE DISTRIBUTIONS

### A.1.1 A MONTE-CARLO APPROACH FOR GENERAL NOISE DISTRIBUTIONS

---

**Algorithm 2** PROBE-MC

---

**Input:** $\mathbf{x}$ s.t. $f(\mathbf{x}) < 0$, $f$, $\sigma^2$, $\lambda > 0$, $t, \alpha, r > 0$
**Init.:** $\mathbf{x}' = \mathbf{x}$;
Compute $\hat{\Delta}_{\mathrm{MC}}(\mathbf{x}')$  ▷ from equation 11
**while** $\hat{\Delta}_{\mathrm{MC}}(\mathbf{x}') > r$ **and** $f(\mathbf{x}') < 0$ **do**
    Compute $\hat{\Delta}_{\mathrm{MC}}(\mathbf{x}')$  ▷ from equation 11
    $\mathbf{x}' = \mathbf{x}' - \alpha \cdot \nabla_{\mathbf{x}'} \mathcal{L}(\mathbf{x}'; \sigma^2, r, \lambda)$
            ▷ Opt. equation 3
**end while**
**Return:** $\check{\mathbf{x}} = \mathbf{x}'$

---

In section 4 we have introduced our PROBE framework, which enables us to guide the search for counterfactual explanations towards regions with a targeted low invalidation rate. Recall that the optimization procedure in Section 4 relied on a first-order approximation to the recourse invalidation rate under Gaussian distributed noisy human responses. In this section, we develop an algorithm that is agnostic to the specifics of the parameterized noise distribution. To this end, we suggest a Monte Carlo estimator of the recourse IR from Def. 1, i.e.,

$$\tilde{\Delta}_{\mathrm{MC}} = \frac{1}{K} \sum_{k=1}^{K} \big(1 - h(\mathbf{x}' + \varepsilon_k)\big). \qquad (8)$$

We highlight that the estimator $\tilde{\Delta}_{\mathrm{MC}}$ allows for a flexible specification of various noise distributions, and thus does not depend on specific distributional assumptions of $\varepsilon$. The following result suggests that we can estimate the true IR $\Delta(\mathbf{x}')$ to desired precision using the Monte-Carlo estimator $\tilde{\Delta}_{\mathrm{MC}}(\mathbf{x}')$.

**Proposition 5.** *The mean-squared-error (MSE) between the true IR $\Delta(\mathbf{x}')$ and the empirical Monte-Carlo estimate $\tilde{\Delta}_{MC}(\mathbf{x}')$ is upper bounded such that:*

$$\mathbb{E}_{\boldsymbol{\varepsilon}}\big[(\Delta(\mathbf{x}') - \tilde{\Delta}_{MC}(\mathbf{x}'))^2\big] \leq \frac{1}{4K}. \qquad (9)$$

Since it is up to us to choose $K$, we can make the MSE arbitrarily small and reliably estimate the true invalidation rate $\Delta(\mathbf{x}')$.

### A.1.2 A DIFFERENTIABLE APPROXIMATION TO $\tilde{\Delta}_{\mathrm{MC}}$

A problem with the estimator $\tilde{\Delta}_{\mathrm{MC}}$ is that it is not amenable to automatic differentiation required for our gradient based algorithm to operate. This is due to the discontinuity at the threshold $\xi$ introduced by the indicator function which, in turn, is applied to the logit score when computing the recourse invalidation rate (i.e., $h(\mathbf{x}) = \mathbb{I}(f(\mathbf{x}) > \xi)$ and see Definition 1). To mitigate this issue, we suggest to use a sigmoid function with appropriate temperature $t$ to approximate the indicator at the threshold $\xi$:

$$S\big((x - \xi) \cdot t\big) = \frac{1}{1 + \exp\big(-(x-\xi) \cdot t\big)}. \qquad (10)$$

Therefore, as $t \to \infty$ the sigmoid $S$ converges

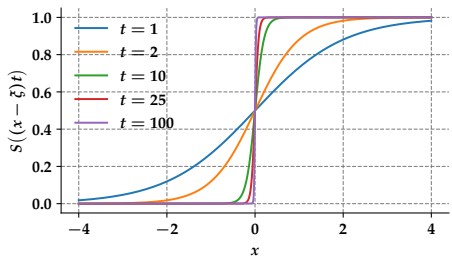

Figure 6: Differentiable approximations of the indicator function $\mathbb{I}(x > 0)$ using the sigmoid function $S(y) = \frac{1}{1+\exp(-y)}$ evaluated at different temperatures $t \in \{1, 2, 10, 25, 100\}$ when $\xi = 0$.

to the indicator function $\mathbb{I}(x > \xi)$. We illustrate this behaviour in Figure 6 for different temperature levels $t \in \{1, 2, 10, 25, 100\}$ when the threshold is $\xi = 0$. Using the differentiable approximation to the indicator function, we are now ready to state a differentiable estimator for the recourse invalidation rate, which we can use to guide our gradient descent procedure to low recourse invalidation regions:

$$\hat{\Delta}_{\mathrm{MC}}(\mathbf{x}'; 0, t) = \frac{1}{K} \sum_{k=1}^{K} \bigg(1 - S\big(t \cdot f(\mathbf{x}' + \varepsilon_k)\big)\bigg). \qquad (11)$$

## A.2 EXTENSIONS TO TREE BASED CLASSIFIERS

The recourse literature commonly considers consequential decision problems which heavily rely on the usage of tabular data. For this data modality, ensembles of decision trees such as Random Forest (RF) (Breiman, 2001) or Gradient Boosted Boosted Decision Trees (GBDT) (Friedman, 2001) are considered among the state-of-the-art models (Borisov et al., 2021). As a consequence, some recourse methods were developed to find recourses for tree ensembles (Tolomei et al., 2017; Lucic et al., 2022) where the non-differentiability prevents a direct application of the recourse objective in equation 1. To extend our method to tree-based classifiers, we also derive an IR expression for tree ensembles, and develop a method which computes low IR recourses for these models.

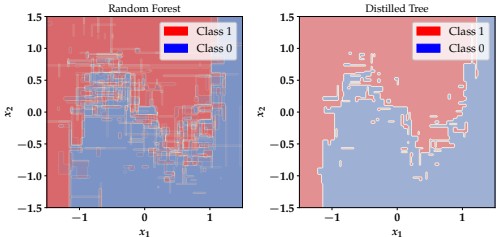
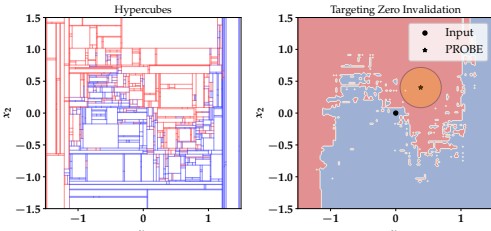

(a) Distilling a RF classifier (left) into a single tree (right). In the left panel, the RF classifier averages 30 decision trees, indicating that the final axis-aligned regions (not shown) are complicated functions of all 30 decision trees.

(b) Computing recourse for the RF model (right) based on the hypercubes (left). The circle has radius $2\sigma$, i.e., it shows the region where 95% of recourse inaccuracies fall when $\sigma^2 = 0.025$. The input $\mathbf{x}$ has IR $\approx 0.5$. The CE has IR $\approx 0.05$.

Figure 7: Computing certified recourses on the 2d Moon data set (Pedregosa et al., 2011) for a RF classifier. **Figure a)**: Distilling a RF classifier (left panel) into a single decision tree (right panel) using knowledge distillation (Domingos, 1997). **Figure b)**: Using the distilled tree, we form the hypercubes (left panel) required to compute IR according to Theorem 2. We then optimize equation 3 to find certified recourses for the RF model (right panel).

**Tree Ensemble Classifiers**    An object of interest is the predicted output of a decision tree:

$$\mathcal{T}(\mathbf{x}) = \sum_{R \in \mathcal{R}_\mathcal{T}} c_\mathcal{T}(R) \cdot \mathbb{I}(\mathbf{x} \in R), \tag{12}$$

where $c_\mathcal{T}(R) \in \{0, 1\}$ is the constant prediction assigned in region $R \in \mathcal{R}_\mathcal{T}$ for tree $\mathcal{T}$. Moreover, a decision forest is formed by a set of $M_T$ decision trees, and forms the probabilistic output:

$$f_{\text{Forest}}(\mathbf{x}) = \frac{1}{M_T} \sum_{m=1}^{M_T} \mathcal{T}_m(\mathbf{x}). \tag{13}$$

The predicted class of an input $\mathbf{x}$ is formed via a vote by the trees where each tree assigns a probability estimate to the input. That is, the predicted class is the one with highest mean probability estimate across the trees. After the trees are combined, the multiple models form a single model again (Domingos, 1997). Thus, the corresponding predicted class of equation 13 is given by:

$$\mathcal{F}(\mathbf{x}) = \sum_{R \in \mathcal{R}_\mathcal{F}} c_\mathcal{F}(R) \cdot \mathbb{I}(\mathbf{x} \in R), \tag{14}$$

where $c_\mathcal{F}(R) \in \{0, 1\}$ is the constant prediction assigned in region $R \in \mathcal{R}_\mathcal{F}$ for the ensemble of trees $\mathcal{F}$. Furthermore, note that for each ensemble, there is an active subset of ensemble-specific features $\mathcal{S}_\mathcal{F} \subseteq \{1, \ldots, d\}$ on which axis-aligned splits took place. Finally, we note that this formulation is quite general as it subsumes a large class of popular tree-based models such as Random Forests (RF) and Gradient Boosted Decision Trees (GBDT).

## A.3 THE RECOURSE IR FOR TREE ENSEMBLE CLASSIFIERS

**Theorem 2** (IR for Tree-Ensemble Classifiers). *Consider the decision forest classifier in equation 14. The recourse invalidation rate under Gaussian distributed response inconsistencies $\varepsilon \sim \mathcal{N}(\mathbf{0}, \sigma^2 \mathbf{I})$ is*

*given by:*

$$\Delta(\check{\mathbf{x}}_E; \boldsymbol{\Sigma}) = 1 - \sum_{R \in \mathcal{R}_{\mathcal{F}}} c_{\mathcal{F}}(R) \prod_{j \in \mathcal{S}_{\mathcal{F}}} d_{j,R}(\check{x}_{E,j}), \tag{15}$$

*where*

$$d_{j,R}(\check{x}_{E,j}) = \left[ \Phi\left( \frac{\bar{t}_{j,R} - \check{x}_{E,j}}{\sigma_j} \right) - \Phi\left( \frac{\underline{t}_{j,R} - \check{x}_{E,j}}{\sigma_j} \right) \right], \tag{16}$$

*and where $\Phi$ is the Gaussian CDF, $\bar{t}_{j,R}$ and $\underline{t}_{j,R}$ are the upper and lower points corresponding to feature $j \in \mathcal{S}_{\mathcal{F}}$ that define the hypercube formed by region $R$.*

*Proof Sketch.* The proof uses the insight that a decision forest based on trees with axis-aligned splits partions the input space into hypercubes where the prediction is either $0$ or $1$. It then remains to evaluate Gaussian integrals subject to the constrains set by the hypercubes. The full proof is given in Appendix D.3. $\qquad\square$

Our proof of Theorem 2 assumed that the split points $\bar{t}_{j,R}$ and $\underline{t}_{j,R}$, corresponding to the tree-ensemble, are readily available. However, the hypercubes formed by the tree-ensemble, for which the prediction is constant, is a function of all individual trees, and of how they are combined. Thus, the clear-cut division into hypecrubes present in each of the trees got lost in the process of model averaging.

**Model Distillation to Evaluate IR**    We suggest a solution to this problem by using a technique called *model distillation* (Domingos, 1997; Bucilua et al., 2006; Hinton et al., 2015; Phuong & Lampert, 2019). In a nutshell: We wish to change the form of the model (to a simpler decision tree) while keeping the same knowledge (from our tree ensemble) (Hinton et al., 2015). Thus, the goal of this technique is to distil the knowledge of a larger model (possibly an ensemble) into a single, small (and interpretable) model. In our case, the ensemble is formed by decision trees, and the target model is a decision tree as well. Second, the method is simple to operationalize: let $h$ be your complex model, and $g$ denotes the simple model. Then we use our data $\{\mathbf{x}_i, y_i\}_{i=1}^n$ to train and validate the model $h$. The target model, however, is trained on samples from $\{\mathbf{x}_i, h(\mathbf{x}_i)\}_{i=1}^n$ to mimic the behaviour of the complex model. We refer to panels 1 to 3 in Figure 7 to gain some intuition on how this technique works on a non-linear 2-dimensional data set.

## B   EXPERIMENTAL DETAILS

In this section, we describe the hyperparameter choices and how the classification models were fitted. We have used CARLA's built-in functionality to fit classifiers using PyTorch (Paszke et al., 2019) and treat all variables as continuous. We set $\lambda_1 = 2$, $\lambda_2 = 1$ and search over $\lambda_3$ in the usual way (Wachter et al., 2018). All models use a $80 - 20$ train-test split for model training and evaluation. We evaluate model quality based on the model accuracy. All models are trained with the same architectures across the data sets:

|  | Neural Network | Logistic Regression |
|---|---|---|
| Units | [Input dim., 50, 2] | [Input dim. , 2] |
| Type | Fully connected | Fully connected |
| Intermediate activations | ReLU | N/A |
| Last layer activations | Softmax | Softmax |

Table 2: Classification model details

|  |  | Adult | COMPAS | Give Me Credit |
|---|---|---|---|---|
| Batch-size | NN | 512 | 32 | 64 |
|  | Logistic Regression | 512 | 32 | 64 |
| Epochs | NN | 50 | 40 | 30 |
|  | Logistic Regression | 50 | 40 | 30 |
| Learning rate | NN | 0.002 | 0.002 | 0.001 |
|  | Logistic Regression | 0.002 | 0.002 | 0.001 |

Table 3: Training details

|  | Adult | COMPAS | Give Me Credit |
|---|---|---|---|
| Logistic Regression | 0.83 | 0.84 | 0.92 |
| Neural Network | 0.85 | 0.85 | 0.93 |

Table 4: Performance of models used for generating recourses

# C  ADDITIONAL EXPERIMENTS

## C.1  ALGORITHMIC RECOURSE IN THE FACE OF NOISY HUMAN RESPONSES

In this Section we show a set of additional experiments. Since this work is the first to highlight and address the problem of recourse invalidation in the face of noisy human responses, we demonstrate in Figures 8 and 9 that recourses generated by state-of-the-art approaches are, on average, invalidated up to 50% of the time when small changes are made to them. It is worth highlighting that the maximum invalidation scores can become as high as 61%, which motivates the need for a recourse method that rightly controls the invalidation rate.

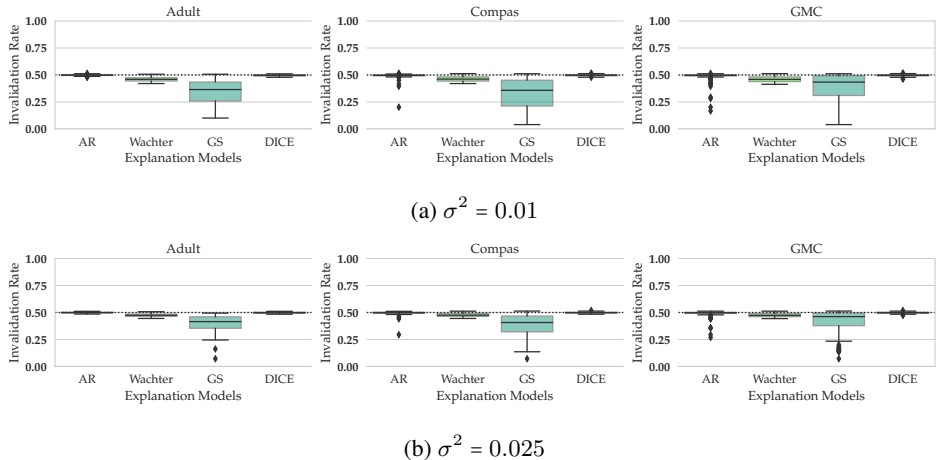

Figure 8: Boxplots of recourse invalidation probabilities across sucessfully generated recourses $\check{\mathbf{x}}$ for **logistic regression** on three data sets. Recourses were generated by four different explanation methods (i.e., AR, Wachter, and GS, DICE), which use different techniques (i.e., integer programming, gradient search, random search, diverse recourse) to find minimum cost recourses. We perturbed the recourses by adding small normally distributed response inaccuracies $\varepsilon \sim \mathcal{N}(\mathbf{0}, \sigma^2 \cdot \mathbf{I})$ to $\check{\mathbf{x}}$.

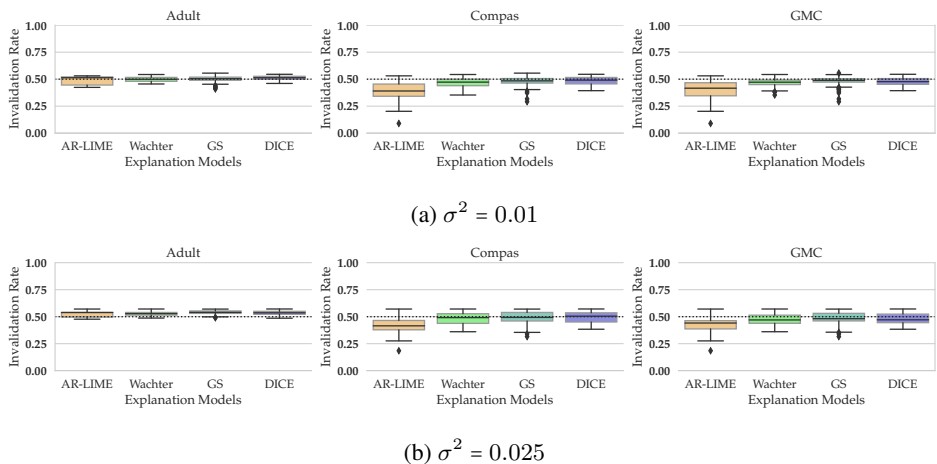

(a) $\sigma^2 = 0.01$

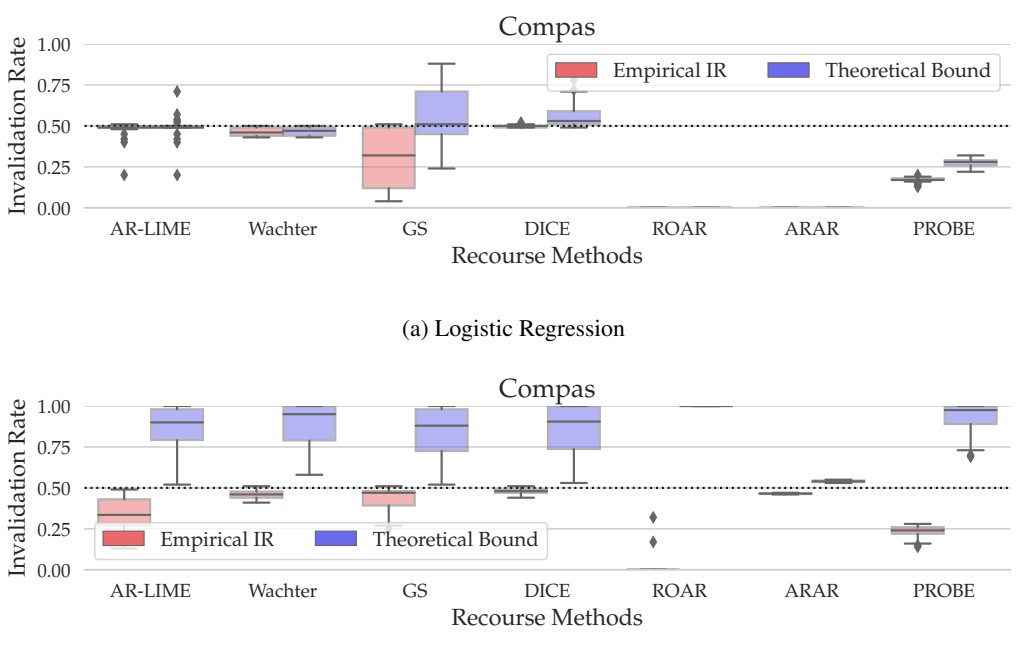

(b) $\sigma^2 = 0.025$

Figure 9: Boxplots of recourse invalidation probabilities across sucessfully generated recourses $\check{x}$ for **NN classifiers** on three data sets. The recourses were generated by four different explanation methods (i.e., AR, Wachter, and GS, DICE), which use different techniques (i.e., integer programming, gradient search, random search, diverse recourse) to find minimum cost recourses. We perturbed the recourses by adding small normally distributed response inaccuracies $\varepsilon \sim \mathcal{N}(\mathbf{0}, \sigma^2 \cdot \mathbf{I})$ to $\check{x}$.

## C.2 MISSING FIGURES FROM THE MAIN TEXT

Below, we show the Figure that was missing from the main text due to space constraints. To keep the plots below more readable, we have omitted DICE from them as both the bounds implied by DICE, the results on cost and the remaining measures are similar to the one by Wachter.

(a) Logistic Regression

(b) Neural Network

Figure 10: Missing figures from the main text (Compas).

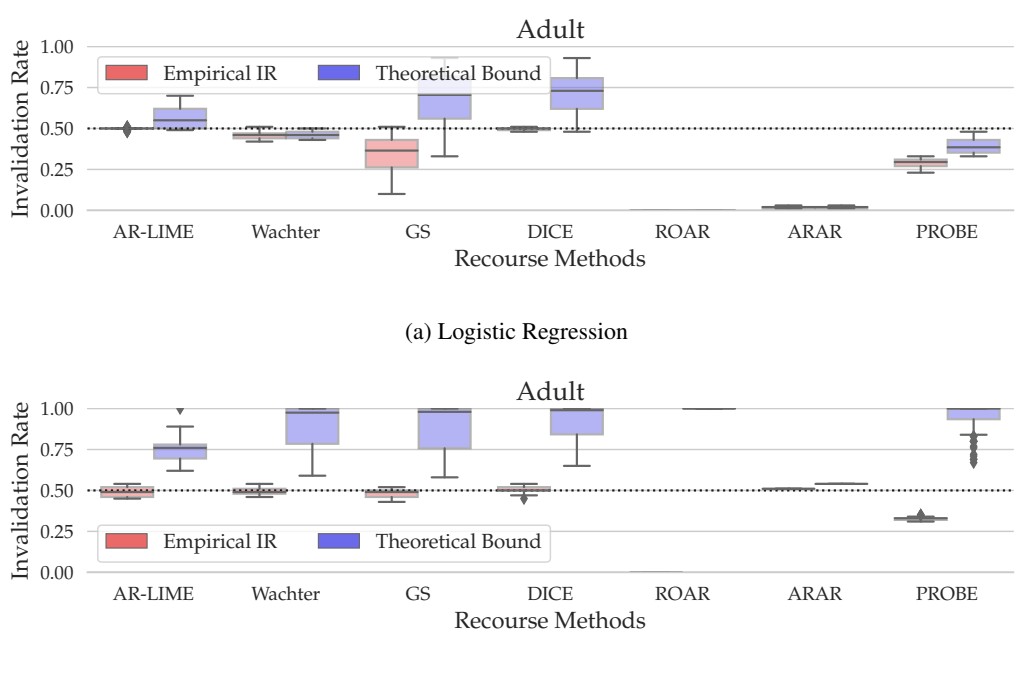

Figure 11: Missing figures from the main text (Adult).

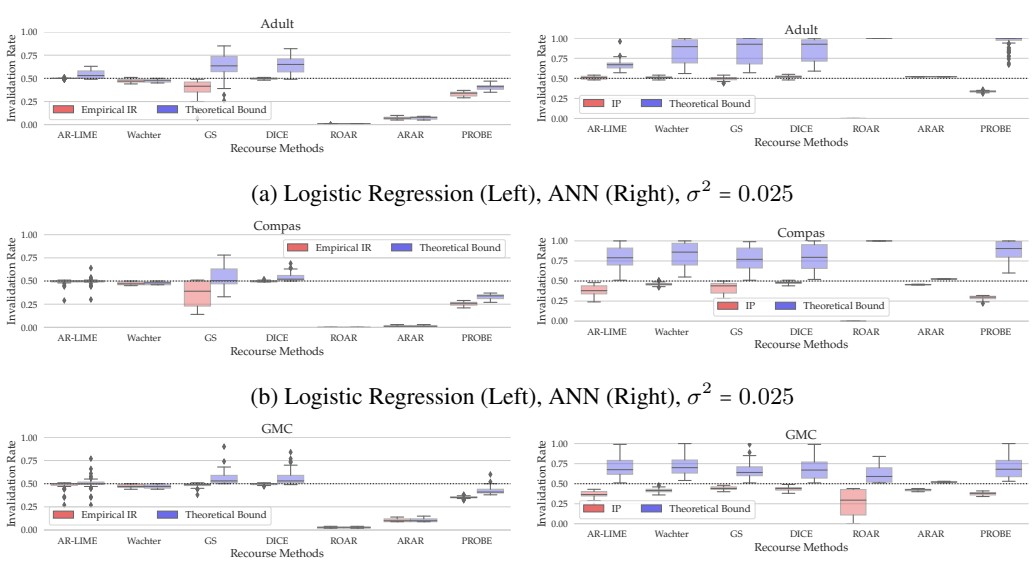

Figure 12: Verifying the theoretical upper bound from Lemma 4 for the logistic regression and artificial neural network classifiers on all data sets when $\sigma^2 = 0.025$. The green boxplots show the empirical recourse IRs for `AR(-LIME)`, `Wachter`, `GS`, and `PROBE`. The blue boxplots show the distribution of upper bounds, which we evaluated by plugging in the corresponding quantities (i.e., $\sigma^2$, $\omega$, etc.) into the upper bound from Lemma 4. The results show no violations of our bounds.

| | | Adult | | | | Compas | | | | GMC | | | |
|---|---|---|---|---|---|---|---|---|---|---|---|---|---|
| | | AR | Wachter | GS | PROBE | AR | Wachter | GS | PROBE | AR | Wachter | GS | PROBE |
| LR | RA (↑) | 0.98 | 1.0 | 1.0 | 1.0 | 1.0 | 1.0 | 1.0 | 1.0 | 1.0 | 1.0 | 1.0 | 1.0 |
| | AIR (↓) | 0.5 ± 0.01 | 0.48 ± 0.01 | 0.4 ± 0.08 | 0.28 ± 0.02 | 0.49 ± 0.03 | 0.48 ± 0.02 | 0.36 ± 0.14 | 0.31 ± 0.01 | 0.48 ± 0.04 | 0.47 ± 0.02 | 0.49 ± 0.02 | 0.3 ± 0.01 |
| | AC (↓) | 0.55 ± 0.4 | 0.62 ± 0.43 | 2.06 ± 1.03 | 2.21 ± 3.17 | 0.16 ± 0.17 | 0.22 ± 0.17 | 0.73 ± 0.45 | 0.68 ± 0.28 | 0.29 ± 0.27 | 0.49 ± 0.51 | 0.28 ± 0.32 | 1.22 ± 2.29 |
| NN | RA (↑) | 0.38 | 1.0 | 1.0 | 1.0 | 0.84 | 1.0 | 1.0 | 1.0 | 0.4 | 1.0 | 1.0 | 1.0 |
| | AIR (↓) | 0.51 ± 0.02 | 0.51 ± 0.01 | 0.5 ± 0.02 | 0.33 ± 0.01 | 0.39 ± 0.06 | 0.46 ± 0.02 | 0.41 ± 0.07 | 0.25 ± 0.02 | 0.37 ± 0.05 | 0.42 ± 0.03 | 0.44 ± 0.02 | 0.34 ± 0.02 |
| | AC (↓) | 1.05 ± 0.22 | 0.3 ± 0.19 | 3.11 ± 1.62 | 1.98 ± 2.35 | 1.15 ± 0.52 | 0.2 ± 0.16 | 1.0 ± 0.17 | 0.84 ± 0.34 | 0.2 ± 0.16 | 0.26 ± 0.18 | 0.11 ± 0.09 | 0.41 ± 0.23 |

Table 5: Recourse accuracy (RA), average recourse invalidation rate (AIR) for $\sigma^2 = 0.025$ and average cost (AC) across different recourse methods. Recourses that use our framework PROBE are more robust compared to those produced by existing baselines. For PROBE, we generated recourses by setting $r = 0.35$. Thus, the AIR should be at most $0.35$, in line with our results.

## C.3 VERIFYING THE VALIDITY OF THE EMPIRICAL INVALIDATION RATE

In Figures 13, 14, and 15 we show that the IRs of the recourses by our framework can be controlled setting $r$ to desired values.

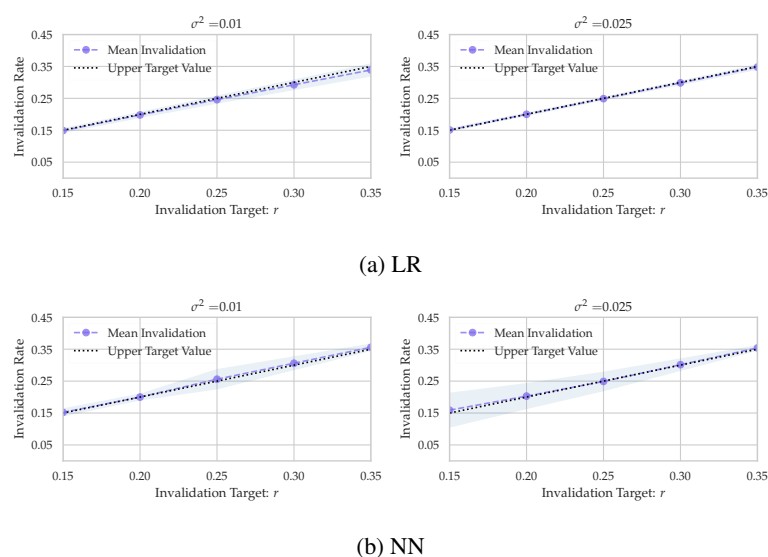

(a) LR

(b) NN

Figure 13: Verifying that the invalidation rate for our framework PROBE (blue line) is at most equal to the invalidation target $r$ on the **Adult** data set for different $\sigma^2 \in \{0.01, 0.025\}$ across both classifiers. We compute the *mean IR* across every instance in the test set. To do that, we sample 10,000 points from $\varepsilon \sim \mathcal{N}(\mathbf{0}, \sigma^2 \mathbf{I})$ for every instance and compute IR in equation 2. Then the *mean IR* quantifies recourse robustness where the individual IRs are averaged over all instances from the test set. The shaded regions indicate the corresponding standard deviations.

## C.4 DEMONSTRATING THE COST-ROBUSTNESS TRADEOFF

In Figures 16 and 17 we demonstrate that there exists a tradeoff between recourse costs and the robustness of recourse to noisy response.

## C.5 DETAILED COMPARISON WITH ROAR AND ARAR

In this section we compare our method with two approaches that aim at generating robust algorithmic recourse in different settings. We further report results by DICE, which does not generate robust recourse. Thus, we PROBE the cost performance (i.e., AC) by DICE to serve as a lower bound, while its robustness performance would serve as an upper bound (i.e., AIR). Regarding the methods that suggest robust recourse we refer to Upadhyay et al. (2021) who proposed a minimax objective to generate recourses that are robust to model updates (ROAR), while Dominguez-Olmedo et al. (2022) use a slight variation of this objective to find recourses that are robust to uncertainty in the

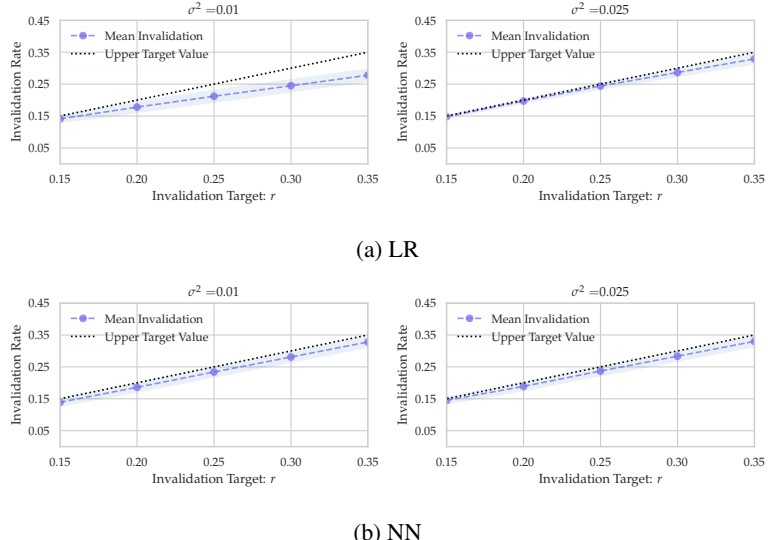

(a) LR

(b) NN

Figure 14: Verifying that the invalidation rate for our framework PROBE (blue line) is at most equal to the invalidation target $r$ on the **Compas** data set for different $\sigma^2 \in \{0.01, 0.025\}$ across both classifiers. We compute the *mean IR* across every instance in the test set. To do that, we sample 10,000 points from $\varepsilon \sim \mathcal{N}(\mathbf{0}, \sigma^2 \mathbf{I})$ for every instance and compute IR in equation 2. Then the *mean IR* quantifies recourse robustness where the individual IRs are averaged over all instances from the test set. The shaded regions indicate the corresponding standard deviations.

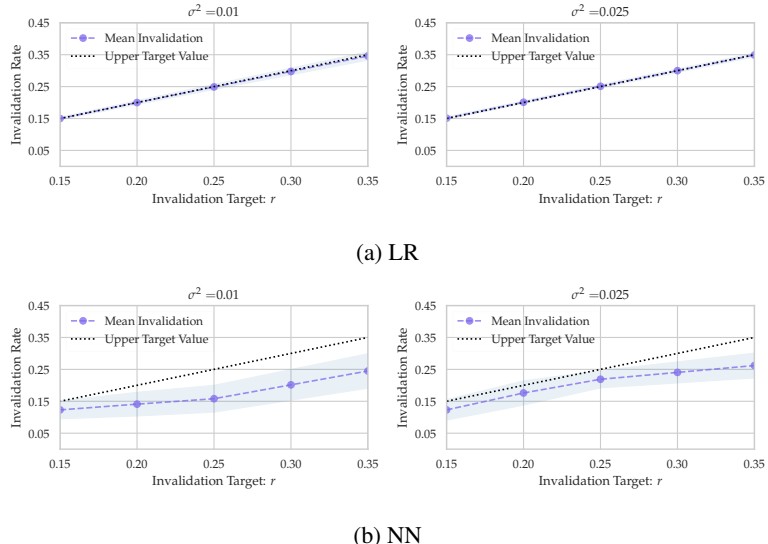

(a) LR

(b) NN

Figure 15: Verifying that the invalidation rate for our framework PROBE (blue line) is at most equal to the invalidation target $r$ on the **GMC** data set for different $\sigma^2 \in \{0.01, 0.025\}$ across both classifiers. We compute the *mean IR* across every instance in the test set. To do that, we sample 10,000 points from $\varepsilon \sim \mathcal{N}(\mathbf{0}, \sigma^2 \mathbf{I})$ for every instance and compute IR in equation 2. Then the *mean IR* quantifies recourse robustness where the individual IRs are averaged over all instances from the test set. The shaded regions indicate the corresponding standard deviations.

inputs (ARAR). Moreover, on a high-level, these objectives differ from our approach since the epsilon neighborhoods that PROBE constructs are probabilistic.

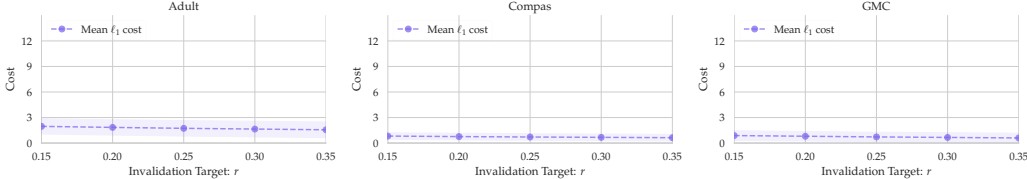

Figure 16: Trading off recourse costs against robustness by choosing the invalidation target $r$ in our PROBE framework. We generated recourses by setting $r \in \{0.20, 0.25, 0.30, 0.35.0.40\}$ and $\sigma^2 = 0.01$ for the **logistic regression classifier**.

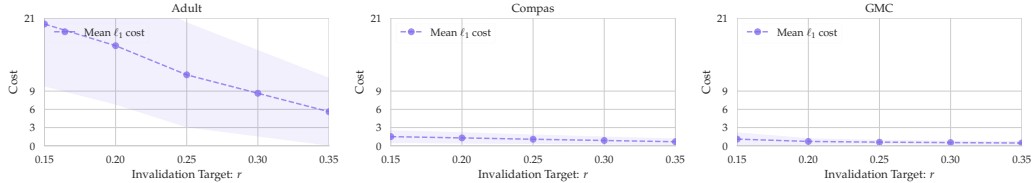

Figure 17: Trading off recourse costs against robustness by choosing the invalidation target $r$ in our PROBE framework. We generated recourses by setting $r \in \{0.20, 0.25, 0.30, 0.35.0.40\}$ and $\sigma^2 = 0.01$ for the **NN classifier**.

**Cost versus invalidation rate performances.** The table shown below summarizes the performance comparison across the aforementioned methods, and Figures 18 and 19 provide Pareto plots, which demonstrate the tradeoff between the average costs measured in terms of $\ell_1$ norm and the average invalidation rate.

**Discussion.** The AIR for PROBE should be at most $0.35$, in line with our results. For ARAR and ROAR, we should expect AIRs close to 0, which is only the case for the linear classifiers. Additionally, ARAR and ROAR provide recourses with up to 10 times higher cost relative to our method PROBE. Note also that ARAR and ROAR have trouble finding recourses for non-linear classifiers, resulting in RA scores of around 5% in the worst case, while not being able to maintain low invalidation scores. This is likely due to the local linear approximation that needs to be used by these methods. For ARAR, only up to 5 percent of all recourse are found (i.e., it only finds recourse with low cost to the decision boundary), and for those identified recourses the average invalidation rate is close to a random coin flip. In summary, PROBE finds recourses for 100% of the test instances in line with the promise of having an invalidation probability of at most $0.35$, while being substantially less costly than ROAR.

| | | Adult | | | Compas | | | GMC | | |
| --- | --- | --- | --- | --- | --- | --- | --- | --- | --- | --- |
| | Measures | ROAR | ARAR | PROBE | ROAR | ARAR | PROBE | ROAR | ARAR | PROBE |
| | RA(↑) | **1.0** | **1.0** | **1.0** | **1.0** | 0.99 | **1.0** | **1.0** | **1.0** | **1.0** |
| LR | AIR (↓) | $0.0 \pm 0.0$ | $0.02 \pm 0.01$ | $0.34 \pm 0.02$ | $0.0 \pm 0.0$ | $0.0 \pm 0.0$ | $0.33 \pm 0.02$ | $0.0 \pm 0.0$ | $0.35 \pm 0.01$ | $0.24 \pm 0.01$ |
| | AC (↓) | $3.56 \pm 0.8$ | $2.68 \pm 0.79$ | $\mathbf{1.56 \pm 0.92}$ | $2.99 \pm 0.31$ | $1.74 \pm 0.3$ | $\mathbf{0.63 \pm 0.39}$ | $1.74 \pm 0.45$ | $1.27 \pm 0.45$ | $\mathbf{0.60 \pm 0.56}$ |
| | RA(↑) | 0.94 | 0.03 | **0.99** | 0.97 | 0.02 | **1.0** | 0.06 | 0.06 | **1.0** |
| NN | AIR (↓) | $0.0 \pm 0.0$ | $0.51 \pm 0.0$ | $0.35 \pm 0.01$ | $0.01 \pm 0.06$ | $0.46 \pm 0.0$ | $0.33 \pm 0.02$ | $0.3 \pm 0.21$ | $0.45 \pm 0.01$ | $\mathbf{0.25 \pm 0.03}$ |
| | AC (↓) | $19.8 \pm 3.39$ | $0.04 \pm 0.0$ | $1.43 \pm 0.49$ | $6.41 \pm 1.07$ | $0.02 \pm 0.0$ | $0.8 \pm 0.34$ | $0.67 \pm 0.94$ | $0.02 \pm 0.0$ | $\mathbf{0.47 \pm 0.21}$ |

Table 6: Recourse accuracy (RA), average recourse invalidation rate (AIR) for $\sigma^2 = 0.01$ and average cost (AC) across different recourse methods. Recourses that use our framework PROBE provide a strong recourse-robustness tradeoff. For PROBE, we generated recourses by setting $r = 0.35$, $\sigma^2 = 0.01$. For ROAR and ARAR, we generated recourses by setting $\varepsilon = 0.01$.

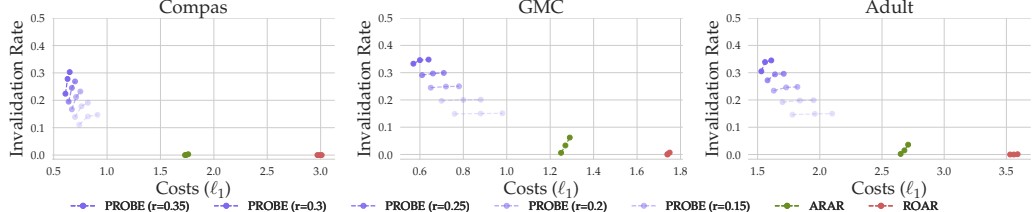

Figure 18: Pareto plots showing the tradeoff between average costs and average invalidation rate when the underlying model is linear. For `PROBE`, the invalidation target $r$ (dotted line) is set to 0.3, and we generated recourses by setting $\sigma^2 \in \{0.005, 0.01, 0.015\}$, and for `ARAR` and `ROAR` we set $\epsilon \in \{0.005, 0.01, 0.015\}$. Following the suggestion by Upadhyay et al. (2021), all recourse methods search for the optimal counterfactuals over the same set of balance parameters $\lambda \in \{0, 0.25, 0.5, 0.75, 1\}$.

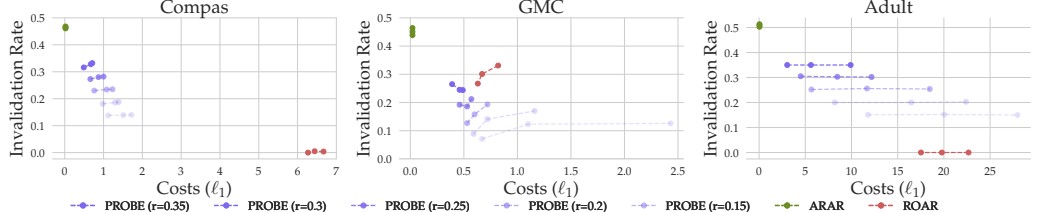

Figure 19: Pareto plots showing the tradeoff between average costs and average invalidation rate when the underlying model is a neural network. For `PROBE`, the invalidation target $r$ (dooted line) is set to 0.35, and we generated recourses by setting $\sigma^2 \in \{0.005, 0.01, 0.015\}$, and for `ARAR` and `ROAR` we set $\epsilon \in \{0.005, 0.01, 0.015\}$. Following the suggestion by Upadhyay et al. (2021), all recourse methods search for the optimal counterfactuals over the same set of balance parameters $\lambda \in \{0, 0.25, 0.5, 0.75, 1\}$.

## D PROOFS

### D.1 PROOF OF PROPOSITION 5

**Proposition 5.** *The mean-squared-error (MSE) between the true IR $\Delta(\mathbf{x}')$ and the empirical Monte-Carlo estimate $\tilde{\Delta}_{MC}(\mathbf{x}')$ is upper bounded such that:*

$$\mathbb{E}_{\boldsymbol{\varepsilon}}\big[(\Delta(\mathbf{x}') - \tilde{\Delta}_{MC}(\mathbf{x}'))^2\big] \leq \frac{1}{4K}. \tag{17}$$

*Proof.* First, recall that the empirical Monte-Carlo estimator is given by:

$$\tilde{\Delta}_{\mathrm{MC}} = \frac{1}{K} \sum_{k=1}^{K} \big(1 - h(\mathbf{x}' + \boldsymbol{\varepsilon}_k)\big). \tag{18}$$

Next, note $\mathbb{E}_{\boldsymbol{\varepsilon}}[1 - h(\mathbf{x}' + \boldsymbol{\varepsilon})] = \Delta(\mathbf{x}')$. Further, the mean-squared error between the nominal invalidation rate $\Delta(\mathbf{x}')$ and the Monte-Carlo estimate $\tilde{\Delta}_{\mathrm{MC}}$ is given by:

$$\mathbb{E}_{\boldsymbol{\varepsilon}}\big[(\Delta(\mathbf{x}') - \tilde{\Delta}_{\mathrm{MC}})^2\big] = \mathbb{V}_{\boldsymbol{\varepsilon}}(\tilde{\Delta}_{\mathrm{MC}}) + \mathbb{E}_{\boldsymbol{\varepsilon}}[\tilde{\Delta}_{\mathrm{MC}} - \Delta(\mathbf{x}')]^2, \tag{19}$$

which gives the bias-variance decomposition. We first compute the squared bias term:

$$\mathbb{E}_{\boldsymbol{\varepsilon}}[\tilde{\Delta}_{\mathrm{MC}} - \Delta(\mathbf{x}')]^2 = \left[\frac{1}{K} \cdot K \cdot \mathbb{E}_{\boldsymbol{\varepsilon}}[1 - h(\mathbf{x} + \boldsymbol{\varepsilon})] - \Delta(\mathbf{x}')\right]^2 \tag{20}$$

$$= 0, \tag{21}$$

where we have used that the $\boldsymbol{\varepsilon}$s are identically distributed. We now turn to the variance term for which we find the following expression:

$$\mathbb{V}_{\boldsymbol{\varepsilon}}(\tilde{\Delta}_{\mathrm{MC}}) = \frac{1}{K^2} \cdot K \cdot \mathbb{V}_{\boldsymbol{\varepsilon}}[1 - h(\mathbf{x} + \boldsymbol{\varepsilon})] = \frac{1}{K} \cdot \mathbb{V}_{\boldsymbol{\varepsilon}}[h(\mathbf{x} + \boldsymbol{\varepsilon})]. \tag{22}$$

It remains to identify an upper bound for $\mathbb{V}_\varepsilon(h(\mathbf{x} + \varepsilon))$. Since $h(\mathbf{x} + \varepsilon)$ is binary, a simple upper bound is given by:

$$\mathbb{V}_\varepsilon(h(\mathbf{x} + \varepsilon)) \le \frac{1}{4}. \tag{23}$$

Combining the expression for the squared bias and the upper bound on the variance yields the desired result. $\qquad\square$

## D.2 Proofs of Theorem 1, Propositions 1 - 4 and Corollary 1

**Theorem 1.** *A first-order approximation $\tilde{\Delta}$ to the recourse invalidation rate $\Delta$ in equation 2 under a Gaussian distribution $\varepsilon \sim \mathcal{N}(\mathbf{0}, \boldsymbol{\Sigma})$ capturing the noise in human responses is given by:*

$$\tilde{\Delta}(\check{\mathbf{x}}_E; \boldsymbol{\Sigma}) = 1 - \Phi\left(\frac{f(\check{\mathbf{x}}_E)}{\sqrt{\nabla f(\check{\mathbf{x}}_E)^\top \boldsymbol{\Sigma} \nabla f(\check{\mathbf{x}}_E)}}\right), \tag{24}$$

*where $\Phi$ is the CDF of the univariate standard normal distribution $\mathcal{N}(0,1)$, $f(\check{\mathbf{x}}_E)$ denotes the logit score at $\check{\mathbf{x}}_E$ which is the recourse output by a recourse method $E$, and $h(\check{\mathbf{x}}_E) \in \{0, 1\}$.*

*Proof.* Let the random variable $\varepsilon$ follow a multivariate normal distribution, i.e., $\varepsilon \sim \mathcal{N}(\boldsymbol{\mu}, \boldsymbol{\Sigma})$. The following result is a well-known fact: $\mathbf{v}^\top \boldsymbol{\epsilon} \sim \mathcal{N}(\mathbf{v}^\top \boldsymbol{\mu}, \mathbf{v}\boldsymbol{\Sigma}\mathbf{v}^T)$ where $\mathbf{v} \in \mathbb{R}^d$. Let $\mathbf{x}$ denote the input sample for which we wish to find a counterfactual $\check{\mathbf{x}}_E = \mathbf{x} + \boldsymbol{\delta}_E$. Recall from Definition 1 that we have to evaluate:

$$\Delta = \mathbb{E}_\varepsilon\left[\underbrace{h(\check{\mathbf{x}}_E)}_{\text{CE class}} - \underbrace{h(\check{\mathbf{x}}_E + \varepsilon)}_{\text{class after response}}\right]$$
$$= 1 - \mathbb{E}_\varepsilon\left[h(\check{\mathbf{x}}_E + \varepsilon)\right], \tag{25}$$

where we have used that the first term is a constant and evaluates to $1$ by the definition of a counterfactual explanation. It remains to evaluate the expectation: $\mathbb{E}_\varepsilon\left[h(\check{\mathbf{x}}_E + \varepsilon)\right]$. Next, we note that equation 25 can equivalently be expressed in terms of the logit outcomes:

$$\Delta = \mathbb{E}_\varepsilon\left[\underbrace{\mathbb{I}\left[f(\check{\mathbf{x}}_E) > 0\right]}_{\text{CE class}} - \underbrace{\mathbb{I}\left[f(\check{\mathbf{x}}_E + \varepsilon) > 0\right]}_{\text{class after perturbation}}\right] = \left(1 - \mathbb{E}_\varepsilon\left[\mathbb{I}\left[f(\check{\mathbf{x}}_E + \varepsilon) > 0\right]\right]\right). \tag{26}$$

Again, we are interested in the second term, which evaluates to:

$$\mathbb{E}_\varepsilon\left[\mathbb{I}\left[f(\check{\mathbf{x}}_E + \varepsilon) > 0\right]\right] = 0 \cdot \mathbb{P}\left(f(\check{\mathbf{x}}_E + \varepsilon) < 0\right) + 1 \cdot \mathbb{P}\left(f(\check{\mathbf{x}}_E + \varepsilon) > 0\right). \tag{27}$$

Next, consider the first-order Taylor approximation: $f(\check{\mathbf{x}}_E + \varepsilon) \approx f(\check{\mathbf{x}}_E) + \nabla f(\check{\mathbf{x}}_E)^\top \varepsilon$. Hence, we know $\nabla f(\check{\mathbf{x}}_E)^\top \varepsilon$ approximately follows $\mathcal{N}(\mathbf{0}, \nabla f(\check{\mathbf{x}}_E)\boldsymbol{\Sigma}\nabla f(\check{\mathbf{x}}_E)^\top)$. Now, the second term can be computed as follows:

$$\mathbb{P}\left(f(\check{\mathbf{x}}_E + \varepsilon) > 0\right) \approx \mathbb{P}\left(f(\check{\mathbf{x}}_E) > -\nabla f(\check{\mathbf{x}}_E)^\top \varepsilon\right) = \mathbb{P}\left(-f(\check{\mathbf{x}}_E) < \nabla f(\check{\mathbf{x}}_E)^\top \varepsilon\right) \tag{28}$$

$$= 1 - \mathbb{P}\left(-f(\check{\mathbf{x}}_E) > \nabla f(\check{\mathbf{x}}_E)^\top \varepsilon\right) \tag{29}$$

$$= 1 - \mathbb{P}\left(\underbrace{\frac{\nabla f(\check{\mathbf{x}}_E)^\top \varepsilon}{\sqrt{\nabla f(\check{\mathbf{x}}_E)^\top \boldsymbol{\Sigma} \nabla f(\check{\mathbf{x}}_E)}}}_{\text{Mean 0 Gaussian RV}} < \underbrace{-\frac{f(\check{\mathbf{x}}_E)}{\sqrt{\nabla f(\check{\mathbf{x}}_E)^\top \boldsymbol{\Sigma} \nabla f(\check{\mathbf{x}}_E)}}}_{\text{Constant}}\right)$$

$$= 1 - \Phi\left(-\frac{f(\check{\mathbf{x}}_E)}{\sqrt{\nabla f(\check{\mathbf{x}}_E)^\top \boldsymbol{\Sigma} \nabla f(\check{\mathbf{x}}_E)}}\right)$$

$$= \Phi\left(\frac{f(\check{\mathbf{x}}_E)}{\sqrt{\nabla f(\check{\mathbf{x}}_E)^\top \boldsymbol{\Sigma} \nabla f(\check{\mathbf{x}}_E)}}\right), \tag{30}$$

where the last line follows due to symmetry of the standard normal distribution (i.e., $\Phi(-u) = 1 - \Phi(u)$). Putting the pieces together, we have:

$$\mathbb{E}_{\varepsilon}\big[\mathbb{I}\big[f(\check{\mathbf{x}}_E + \varepsilon) > 0\big]\big] = 0 \cdot \mathbb{P}\bigg(f(\check{\mathbf{x}}_E + \varepsilon) < 0\bigg) + 1 \cdot \mathbb{P}\bigg(f(\check{\mathbf{x}}_E + \varepsilon) \geq 0\bigg) \tag{31}$$

$$= \Phi\bigg(\frac{f(\check{\mathbf{x}}_E)}{\sqrt{\nabla f(\check{\mathbf{x}}_E)^\top \boldsymbol{\Sigma} \nabla f(\check{\mathbf{x}}_E)}}\bigg). \tag{32}$$

Thus, we have:

$$\Delta \approx \tilde{\Delta} = 1 - \Phi\bigg(\frac{f(\check{\mathbf{x}}_E)}{\sqrt{\nabla f(\check{\mathbf{x}}_E)^\top \boldsymbol{\Sigma} \nabla f(\check{\mathbf{x}}_E)}}\bigg), \tag{33}$$

which completes our proof. Note that this is equivalent to $\mathbb{P}\bigg(f(\check{\mathbf{x}}_E + \varepsilon) < 0\bigg)$, and thus we are "counting" how often perturbations to $\check{\mathbf{x}}_E$ sampled from $\varepsilon \sim \mathcal{N}(\mathbf{0}, \boldsymbol{\Sigma})$ result in flips back to the undesired class. $\square$

**Proposition 2.** *For a linear classifier, let $r \in (0, 1)$ and let $\check{\mathbf{x}}_E = \mathbf{x} + \boldsymbol{\delta}_E$ be the output produced by some recourse method $E$ such that $h(\check{\mathbf{x}}_E) = 1$. Then the cost required to achieve a fixed invalidation target $r$ is given by:*

$$\|\boldsymbol{\delta}_E\|_2 = \frac{\sigma}{\omega}\big(\Phi^{-1}(1 - r) - c\big), \tag{34}$$

*where $c = \frac{f(\mathbf{x})}{\sigma \cdot \|\nabla f(\mathbf{x})\|_2}$ is a constant, and $\omega > 0$ is the cosine of the angle between the vectors $\nabla f(\mathbf{x})$ and $\boldsymbol{\delta}_E$.*

*Proof.* Under a logistic classifier, the result immediately follows by setting the expression from Theorem 1 equal to $r$, using the identity $\nabla f(\mathbf{x})^\top \boldsymbol{\delta}_E = \omega \cdot \|\nabla f(\mathbf{x})\|_2 \cdot \|\boldsymbol{\delta}_E\|_2$ where $\omega$ is the cosine of the angle between the vectors $\nabla f(\mathbf{x})$ and $\boldsymbol{\delta}_E$, and rearranging for $\|\boldsymbol{\delta}_E\|_2$. $\square$

**Proposition 3.** *Under the same conditions as in Proposition 2, we have $\frac{\partial \|\boldsymbol{\delta}_E\|_2}{\partial(1-r)} = \frac{\sigma}{\omega}\frac{1}{\phi(\Phi^{-1}(1-r))} > 0$, i.e., an infinitesimal increase in robustness (i.e.,$1 - r$) increases the cost of recourse by $\frac{\sigma}{\omega}\frac{1}{\phi(\Phi^{-1}(1-r))}$.*

*Proof.* We will compute the derivative of $\|\boldsymbol{\delta}_E\|_2 = \frac{\sigma}{\omega}\big(\Phi^{-1}(1 - r) - c\big)$ with respect to $1 - r$ and show that it is positive for all $r \in (0, 1)$:

$$\frac{\partial \|\boldsymbol{\delta}_E\|_2}{\partial(1 - r)} = \frac{\sigma}{\omega}\frac{1}{\phi(\Phi^{-1}(1 - r))} > 0, \tag{35}$$

where $\phi$ is the probability density function (PDF) of the standard Gaussian distribution. Since the PDF must be positive, we have that $\phi(\Phi^{-1}(1 - r)) > 0$, and we know that $\sigma, \omega > 0$. Thus, the results follows. $\square$

**Proposition 4.** *Let $\check{\mathbf{x}}_E$ be the output produced by some recourse method $E$ such that $h(\check{\mathbf{x}}_E) = 1$. Then, an upper bound on $\tilde{\Delta}$ from equation 4 is given by:*

$$\tilde{\Delta}(\check{\mathbf{x}}_E; \sigma^2 \mathbf{I}) \leq 1 - \Phi\bigg(c + \frac{\omega}{\sigma}\frac{\|\nabla f(\mathbf{x})\|_2}{\|\nabla f(\check{\mathbf{x}}_E)\|_2}\frac{\|\boldsymbol{\delta}_E\|_1}{\sqrt{\|\boldsymbol{\delta}_E\|_0}}\bigg), \tag{36}$$

*where $c = \frac{f(\mathbf{x})}{\sigma \cdot \|\nabla f(\mathbf{x})\|_2}$ is a constant, $\boldsymbol{\delta}_E = \check{\mathbf{x}}_E - \mathbf{x}$, and $\omega > 0$ is the cosine of the angle between the vectors $\nabla f(\mathbf{x})$ and $\boldsymbol{\delta}_E$.*

*Proof.* We start by noting the following basic inequality:

$$\|\mathbf{z}\|_1 \leq \sqrt{\|\mathbf{z}\|_0} \cdot \|\mathbf{z}\|_2.$$

Going forward, we will refer to these inequalities as basic inequalities. Moreover, note that $\Phi$ is a monotonic function. Thus, we have $\Phi(a) \le \Phi(a')$ for $a \le a'$. Note that $f(\check{\mathbf{x}}_E) \approx f(\mathbf{x}) + \nabla f(\mathbf{x})^\top \boldsymbol{\delta}_E$. Thus we obtain the following approximation:

$$\tilde{\Delta} = 1 - \Phi\left(\frac{f(\mathbf{x}) + \nabla f(\mathbf{x})^\top \boldsymbol{\delta}_E}{\sqrt{\nabla f(\check{\mathbf{x}}_E)\boldsymbol{\Sigma}^\top \nabla f(\check{\mathbf{x}}_E)}}\right). \tag{37}$$

Next, we will find upper bounds for the term on the right: Before we will do that, we will express the above expression more conveniently to highlight the impact of the counterfactual action $\boldsymbol{\delta}_E$ more explicitly. To do that, note that $\nabla f(\mathbf{x})^\top \boldsymbol{\delta}_E = \omega \cdot \|\nabla f(\mathbf{x})\|_2 \cdot \|\boldsymbol{\delta}_E\|_2$ where $\omega$ is the cosine of the angle between the vectors $\nabla f(\mathbf{x})$ and $\boldsymbol{\delta}_E$. Using $\boldsymbol{\Sigma} = \sigma^2 \mathbf{I}$, we obtain:

$$\Phi\left(\frac{f(\mathbf{x}) + \nabla f(\mathbf{x})^\top \boldsymbol{\delta}_E}{\sigma\|\nabla f(\check{\mathbf{x}}_E)\|_2}\right) = \Phi\left(c + \frac{\|\nabla f(\mathbf{x})\|_2}{\|\nabla f(\check{\mathbf{x}}_E)\|_2} \cdot \frac{\omega}{\sigma} \cdot \|\boldsymbol{\delta}_E\|_2\right), \tag{38}$$

where we defined a constant $c = \frac{f(\mathbf{x})}{\sigma\|\nabla f(\check{\mathbf{x}}_E)\|_2}$ using quantities that we will keep fixed in our analysis, namely $\mathbf{x}, \nabla f(\mathbf{x})$ and $\sigma$. Also note that $\mathbf{x}$ is the factual input, and thus its logit score satisfies: $f(\mathbf{x}) < 0$. Since $\boldsymbol{\delta}_E$ is a valid perturbation, we must have that $\omega > 0$ for the perturbation to change the class prediction.

Note that the following *lower bound* holds by the basic inequality stated above:

$$\Phi\left(c + \frac{\|\nabla f(\mathbf{x})\|_2}{\|\nabla f(\check{\mathbf{x}}_E)\|_2} \cdot \frac{\omega}{\sigma} \cdot \|\boldsymbol{\delta}_E\|_2\right) \ge \Phi\left(c + \frac{\|\nabla f(\mathbf{x})\|_2}{\|\nabla f(\check{\mathbf{x}}_E)\|_2} \cdot \frac{\omega}{\sigma} \cdot \frac{\|\boldsymbol{\delta}_E\|_1}{\sqrt{\|\boldsymbol{\delta}_E\|_0}}\right). \tag{39}$$

As a consequence we obtain the following *upper bound on the IR*:

$$\tilde{\Delta} \le 1 - \Phi\left(c + \frac{\|\nabla f(\mathbf{x})\|_2}{\|\nabla f(\check{\mathbf{x}}_E)\|_2} \cdot \frac{\omega}{\sigma} \cdot \frac{\|\boldsymbol{\delta}_E\|_1}{\sqrt{\|\boldsymbol{\delta}_E\|_0}}\right), \tag{40}$$

as claimed. $\qquad\square$

**Proposition 1.** *For the logistic regression classifier, consider the recourse output by* Wachter et al. *(2018):* $\check{\mathbf{x}}_{Wachter}(s) = \mathbf{x} + \frac{s-f(\mathbf{x})}{\|\nabla f(\mathbf{x})\|_2^2}\nabla f(\mathbf{x})$. *Then the recourse invalidation rate has the following closed-form:*

$$\Delta(\check{\mathbf{x}}_{Wachter}(s); \sigma^2\mathbf{I}) = 1 - \Phi\left(\frac{s}{\sigma\|\nabla f(\mathbf{x})\|_2}\right), \tag{41}$$

*where $s$ is the target logit score.*

*Proof.* Since we are in the linear case, we have: $\nabla f(\check{\mathbf{x}}_E) = \nabla f(\mathbf{x})$. Also, note that $f(\check{\mathbf{x}}_E) = f(\mathbf{x}) + \nabla f(\mathbf{x})^\top \boldsymbol{\delta}_E$. Using $\boldsymbol{\Sigma} = \sigma^2\mathbf{I}$, we obtain the following exact expression:

$$\Delta = 1 - \Phi\left(\frac{f(\mathbf{x}) + \nabla f(\mathbf{x})^\top \boldsymbol{\delta}_E}{\sigma\|\nabla f(\mathbf{x})\|_2}\right). \tag{42}$$

From Pawelczyk et al. (2022), we have:

$$\boldsymbol{\delta}_{\text{Wachter}} = \frac{s - f(\mathbf{x})}{\|\nabla f(\mathbf{x})\|_2^2}\nabla f(\mathbf{x}). \tag{43}$$

Plugging equation 43 into equation 42 we obtain:

$$\Delta = 1 - \Phi\left(\frac{f(\mathbf{x})}{\sigma\|\nabla f(\mathbf{x})\|_2} + \frac{\nabla f(\mathbf{x})^\top \boldsymbol{\delta}_E}{\sigma\|\nabla f(\mathbf{x})\|_2}\right) \tag{44}$$

$$= 1 - \Phi\left(\frac{f(\mathbf{x})}{\sigma\|\nabla f(\mathbf{x})\|_2} + \frac{1}{\sigma\|\nabla f(\mathbf{x})\|_2} \cdot \nabla f(\mathbf{x})^\top \nabla f(\mathbf{x})\frac{s - f(\mathbf{x})}{\|\nabla f(\mathbf{x})\|_2^2}\right)$$

$$= 1 - \Phi\left(\frac{f(\mathbf{x})}{\sigma\|\nabla f(\mathbf{x})\|_2} + \frac{s - f(\mathbf{x})}{\sigma\|\nabla f(\mathbf{x})\|_2}\right)$$

$$= 1 - \Phi\left(\frac{s}{\sigma\|\nabla f(\mathbf{x})\|_2}\right), \tag{45}$$

which concludes the proof. $\qquad\square$

**Corollary 1.** *Under the conditions of Proposition 1, choosing $s_r = \sigma \|\nabla f(\mathbf{x})\|_2 \Phi^{-1}(1-r)$ guarantees a recourse invalidation rate of $r$, i.e., $\Delta(\check{\mathbf{x}}_{Wachter}(s_r); \sigma^2 \mathbf{I}) = r$.*

*Proof.* The result directly follows from plugging in $s_r = \sigma \|\nabla f(\mathbf{x})\|_2 \Phi^{-1}(1-r)$ into the optimal recourse from $\delta_{\text{Wachter}}$ from equation 43 and subsequently evaluating the recourse invalidation rate from equation 5. □

### D.3 PROOF OF THEOREM 2

*Proof.* From Definition 1 we know:

$$\Delta_{\text{Forest}} = \mathbb{E}_{\boldsymbol{\varepsilon}}\left[\underbrace{\mathcal{F}(\check{\mathbf{x}}_E)}_{\text{CE class}} - \underbrace{\mathcal{F}(\check{\mathbf{x}}_E + \boldsymbol{\varepsilon})}_{\text{class after response}}\right] \tag{46}$$

$$= 1 - \mathbb{E}_{\boldsymbol{\varepsilon}}\left[\mathcal{F}(\check{\mathbf{x}}_E + \boldsymbol{\varepsilon})\right]. \tag{47}$$

It remains to evaluate: $\mathbb{E}_{\boldsymbol{\varepsilon}}\left[\mathcal{F}(\check{\mathbf{x}}_E + \boldsymbol{\varepsilon})\right]$. Using equation 14, we have:

$$\mathbb{E}_{\boldsymbol{\varepsilon}}\left[\mathcal{F}(\check{\mathbf{x}}_E + \boldsymbol{\varepsilon})\right] = \mathbb{E}_{\boldsymbol{\varepsilon}}\left[\sum_{R \in \mathcal{R}_{\mathcal{F}}} c_{\mathcal{F}}(R) \cdot \mathbb{I}(\check{\mathbf{x}}_E + \boldsymbol{\varepsilon} \in R)\right] \tag{48}$$

$$= \sum_{R \in \mathcal{R}_{\mathcal{F}}} c_{\mathcal{F}}(R) \cdot \mathbb{E}_{\boldsymbol{\varepsilon}}\left[\mathbb{I}(\check{\mathbf{x}}_E + \boldsymbol{\varepsilon} \in R)\right] \qquad \text{(Linearity of Expectation)}$$

$$= \sum_{R \in \mathcal{R}_{\mathcal{F}}} c_{\mathcal{F}}(R) \cdot \int_R p(\mathbf{y}) d\mathbf{y} \qquad (p(\mathbf{y}) = \mathcal{N}(\check{\mathbf{x}}_E, \boldsymbol{\sigma}^2 \mathbf{I}))$$

$$= \sum_{R \in \mathcal{R}_{\mathcal{F}}} c_{\mathcal{F}}(R) \cdot \prod_{j \in \mathcal{S}_{\mathcal{F}}} \int_{R_j} \frac{1}{\sqrt{2\pi\sigma_j^2}} \exp\left(-\frac{1}{2}\frac{(y_j - \check{x}_{E,j})^2}{\sigma_j^2}\right) dy_j$$

$$\text{(Since } \varepsilon \text{ is an independent Gaussian)}$$

$$= \sum_{R \in \mathcal{R}_{\mathcal{F}}} c_{\mathcal{F}}(R) \cdot \prod_{j \in \mathcal{S}_{\mathcal{F}}} \left[\Phi\left(\frac{\bar{t}_{j,R} - \check{x}_{E,j}}{\sigma_j}\right) - \Phi\left(\frac{\underline{t}_{j,R} - \check{x}_{E,j}}{\sigma_j}\right)\right].$$

$$\text{(Since } \varepsilon \text{ is Gaussian)}$$

Using our Definition of robustness, we have

$$\Delta_{\text{Forest}} = 1 - \sum_{R \in \mathcal{R}_{\mathcal{F}}} c_{\mathcal{F}}(R) \prod_{j \in \mathcal{S}_{\mathcal{F}}} \left[\Phi\left(\frac{\bar{t}_{j,R} - \check{x}_{E,j}}{\sigma_j}\right) - \Phi\left(\frac{\underline{t}_{j,R} - \check{x}_{E,j}}{\sigma_j}\right)\right], \tag{49}$$

which completes the proof. □

