# OpenReview forum: "Probabilistically Robust Recourse: Navigating the Trade-offs between Costs and Robustness in Algorithmic Recourse"
_ICLR.cc/2023/Conference — ICLR 2023 poster_

### Official Review · Reviewer_WYC5 · 2022-10-25

**Confidence:** 3
**Correctness:** 4
**Technical Novelty And Significance:** 2
**Empirical Novelty And Significance:** 4
**Recommendation:** 6

**Clarity, Quality, Novelty And Reproducibility:**

The presentation of the paper can be improved. Proofs and the experimental details in the appendix needs some referencing in the main paper. There are minor typos.

**Strength And Weaknesses:**

Strengths :
1. The area of algorithmic recourse has a great impact and is an important problem to study. The current work addresses an important problem of managing the trade-off between two highly desired objectives (recourse cost and robustness).
2. The definition of invalidation rate is characterized in a closed-form such that PROBE can be efficiently implemented.
3. The empirical evaluation on multiple datasets showed no violation with the theoretical guarantees provided (Figure 5 and Proposition 3)


Weaknesses :
1. The convergence of the algorithm PROBE is not discussed. Since the optimization of the objective in Equation (3) in PROBE as discussed is being done via gradient descent, it would be interesting to compare the convergence of PROBE with existing baselines.
2. The presentation of the paper can be improved in various places like (1) Referencing the proofs in the main paper. (2) Outline the additional experiment results in the appendix.


**Summary Of The Paper:**

1. The paper proposed a new user-driven algorithmic framework PROBE (Probabilistically Robust Recourse) that tackles two independently well-studied challenges in algorithmic recourse  -- (1) recourse cost, and (2) robustness under noisy human implementation. This is the first study where both recourse cost and robustness are simultaneously optimized.

2. PROBE enables the end users to navigate the tradeoffs between recourse cost and robustness via an input parameter $r$ which is a user-defined upper-bound on the invalidation rate i.e., recourse can be invalidated at most $r$ percent of the time. Note that $r=1$ corresponds to high-risk and low recourse cost and $r=0$ is low-risk and high-cost recourse.

3. The main contributions of this paper are (i) the definition of the recourse invalidation rate (IR) in Definition 1 and Equation (2) (ii) approximating the invalidation rates of any given instance w.r.t different underlying prediction models like linear models, non-linear models and tree-based models, etc. Thus, the paper efficiently implements the PROBE algorithm (1) using the closed form of approximated invalidation rates (Theorem 1).

4. There are an extensive set of experiments conducted on multiple datasets comparing PROBE to existing algorithms that minimize recourse cost (Watcher, AR-LIME, DICE, and GS -- Figure 5) and robust algorithms (ROAR, ARAR - Figure 5). Their empirical evaluation using well-defined measures like "recourse accuracy (RA)", "average recourse invalidation rate (AIR)" and "average cost (AC)" show that PROBE generated recourses that are more robust than existing baselines (Table 1). Further, the empirical evaluation supported the theoretical upper bounds on IR in Proposition 3.



**Summary Of The Review:**

I believe the paper has studied an important problem and made strong technical contributions. The empirical evaluation is extensive and the results are inline with the theoretical analysis. The presentation of the paper in my opinion can be improved to help readability.

---

> ### Author Response · Authors · 2022-11-17
> **Response to Reviewer WYC5**
>
> We thank the reviewer for their detailed feedback and are glad to see that the reviewer appreciates the ideas investigated in this work. Below we provide a detailed response to the reviewer.
>
>
> > The convergence of the algorithm PROBE is not discussed. Since the optimization of the objective in Equation (3) in PROBE as discussed is being done via gradient descent, it would be interesting to compare the convergence of PROBE with existing baselines.
>
> We chose to analyze the PROBE objective and the corresponding optimal solution (Proposition 1 and Corollary 1) instead of our optimization procedure because we wanted to first answer the question of whether we can make theoretical claims about our conceptual formulation. Furthermore, our optimization procedure employs gradient descent to solve our optimization problem. This is a popular and already well studied area in machine learning and optimization, and we did not consider providing convergence guarantees about these general purpose algorithms to be within the scope of this work. However, prior research has already demonstrated that using *multiple restarts alongside gradient descent* style approaches provides reasonable solutions to non-convex problems resulting in meaningful recourses [1,2]. Given all this, we chose to analyze the PROBE objective and the corresponding optimal solution instead of our optimization procedure. Analyzing Algorithm 1 carefully and improving the optimization framework is certainly important future work.
>
> Additionally, when the underlying classifier is linear or can be approximated by a local linear function we can use the insights from Proposition 1 in the main paper to provide convergence guarantees: according to this Proposition we can adjust the logit score $s$ featured in the closed-form recourse solution [4] from the recourse objective in Wachter et al [3] to yield a recourse with guaranteed target invalidation rate $r$. To make this more clear, we have added a corollary below Proposition 1.
>
> In Figures 4a and 4b we do implicitly analyze the convergence properties of our algorithm relative to state of the art robust recourse generating methods where convergence here refers to the empirical invalidation rate being smaller than $r$. As desired, we find that our algorithm converges to invalidation rates that lie below the targeted invalidation rate $r$, while the other state of the art methods do not converge to the targeted invalidation rate of $r=0$.
>
> Finally, in Appendix C.3 we empirically verify that the empirical invalidation rate by PROBE is at most equal to the targeted invalidation rate $r$. This finding is consistent across all data sets, which we verify for a set of reasonable values of $r \in$ {0.15, 0.20, 0.25, 0.30, 0.35}.
>
> > The presentation of the paper can be improved in various places like (1) Referencing the proofs in the main paper. (2) Outline the additional experiment results in the appendix.
>
> Thank you very much for your constructive suggestions. Following your suggestions, we have (i) added additional references to the proofs in section 4, and (ii) we have added a short discussion in Section 5 on the experimental results that were relegated to Appendix C.
>
> ---
> **References**
>
> [1] Dick et al., “How many random restarts are enough”. Technical report, 2014.
>
> [2] Jason D. Lee , Max Simchowitz , Michael I. Jordan, and Benjamin Recht, “Gradient Descent Converges to Minimizers”, 9th Annual Conference on Learning Theory, PMLR 49:1246-1257, 2016.
>
> [3] Wachter et al., "Counterfactual explanations without opening the black box: Automated decisions and the GDPR." Harv. JL & Tech., 2017.
>
> [4] Martin Pawelczyk, Chirag Agarwal, Shalmali Joshi, Sohini Upadhyay, and Himabindu Lakkaraju.Exploring counterfactual explanations through the lens of adversarial examples: A theoretical and empirical analysis. In International Conference on Artificial Intelligence and Statistics (AISTATS), 2022

---

### Official Review · Reviewer_ifQy · 2022-10-26

**Confidence:** 3
**Correctness:** 4
**Technical Novelty And Significance:** 2
**Empirical Novelty And Significance:** 3
**Recommendation:** 8

**Clarity, Quality, Novelty And Reproducibility:**

The framework proposed by this paper is novel, in the sense that it studies the question of robust low-cost recourse mechanisms under noisy implementations. The problem and applications are well motivated in the paper. Relevant examples and intuition specified whenever applicable help better understand the subject matter. Overall, it is a well-written paper.

**Strength And Weaknesses:**

Strengths:
The main contribution of this paper is the notion of IR and how to find a good approximation to it efficiently; and the modified objective function to incorporate IR such that the output recourse not only has low cost but also low invalidation probability. I believe IR and the modified objective are of interest for developing low-cost robust recourse mechanisms. The extensive experimental results support the claims of the paper well.

Weaknesses:
1) How were the r values chosen for each dataset? A small description of this would be useful.
2) Are there any convergence guarantees for PROBE? Is it guaranteed that the IR will be less than r eventually?

**Summary Of The Paper:**

In this paper, the authors give a framework, called PROBE (Probabilistically Robust Recourse), to address the problem of providing low cost recourse that is also robust. Prior work in this context focused on either achieving low recourse costs, that led to mechanisms that were very sensitive to input perturbations or small shifts in the prediction model considered; or mechanisms that were highly robust, but incur high recourse costs. Further, in real life, recourse mechanisms are implemented noisily, thus, there is a need to manage the trade-offs between recourse cost and robustness effectively. The authors address the latter in this paper. Towards this, they define the notion of invalidation rate (IR), which denotes the probability with which a recourse gets invalidated due to small changes in the recourse arising due to human error. A threshold for IR is chosen by users to navigate trade offs between cost and robustness. The authors propose a new objective, that adds a Hinge loss term along with the standard objective function to ensure that the output recourse has a low probability of being invalidated. The main algorithm proposed essentially performs a gradient descent, while considering an approximation to the value of IR at each time step. The authors also give general upper bounds for the value of IR given any recourse method. Finally, an extensive empirical analysis is performed, comparing costs and IR for PROBE and other existing frameworks.

**Summary Of The Review:**

Overall, I believe the contributions of this work are interesting to the Algorithmic Recourse community. The proposed framework is simple, cost effective, and powerful in terms of allowing users to choose the trade off parameter r. Following are a few line-by-line comments.

Minor Comments:
- Page 3: para 2, end of line 5: should be minimax
- Page 3: equation (1), remove extra ‘)’ in term 1 of rhs
- Page 4: equation (3), should it be R(x’; r , \sigma^2 I)? Also, remove extra ‘)’ in the second term.
- Page 4: last para, line 3, a output -> an output
- Page 5: Sec 4.2 para 1, line 5, Section not referenced
- Page 9: para 1, line 2, recousre -> recourse

---

> ### Author Response · Authors · 2022-11-17
> **Response to Reviewer ifQy**
>
> We thank the reviewer for their insightful feedback and are very grateful that the reviewer appreciates the novelty and significance of our work. Please find our response below.
>
> >How were the $r$ values chosen for each dataset? A small description of this would be useful.
>
> Thank you for your suggestion. Recall that a value of $r=1$ is not useful as it implies that all recourses would be invalidated. At the other extreme, low values of $r$ lead to very high recourse costs (see Figure 4a and 4b). Moreover, a value of $r=0.5$ is equivalent to a random coin toss. For example, Figures 8 and 9 from Appendix C show that empirical invalidation rates are usually close to 0.5 for baseline state of the art methods. So, we considered a moderate value of 0.35 as a reasonable compromise in our experiments in the main paper.
> Also, note that we experiment with a wide range of $r$ values in figures 4a and 4b. These figures showcase cost and invalidation rate results of a wide spectrum of reasonable choices for $r \in$ {0.35, 0.30, 0.25, 0.20, 0.15 }.
>
> > Are there any convergence guarantees for PROBE? Is it guaranteed that the IR will be less than $r$ eventually?
>
> We chose to analyze the PROBE objective and the corresponding optimal solution (Proposition 1 and Corollary 1) instead of our optimization procedure because we wanted to first answer the question of whether we can make theoretical claims about our conceptual formulation. Furthermore, our optimization procedure employs gradient descent to solve our optimization problem. This is a popular and already well studied area in machine learning and optimization, and we did not consider providing convergence guarantees about these general purpose algorithms to be within the scope of this work. However, prior research has already demonstrated that using *multiple restarts alongside gradient descent* style approaches provides reasonable solutions to non-convex problems resulting in meaningful recourses [1,2]. Given all this, we chose to analyze the PROBE objective and the corresponding optimal solution instead of our optimization procedure. Analyzing Algorithm 1 carefully and improving the optimization framework is certainly important future work.
>
> Additionally, when the underlying classifier is linear or can be approximated by a locally linear function we can use the insights from Proposition 1 in the main paper to provide convergence guarantees: according to this Proposition we can adjust the logit score $s$ featured in the closed-form recourse solution [4] from the recourse objective in Wachter et al [3] to yield a recourse with guaranteed target invalidation rate $r$. To make this more clear, we have added a corollary below Proposition 1.
>
> Finally, in Appendix C.3 we empirically verify that the empirical invalidation rate by PROBE is at most equal to the targeted invalidation rate $r$. This finding is consistent across all data sets, which we verify for a set of reasonable values of $r \in $ {0.15, 0.20, 0.25, 0.30, 0.35}.
>
> ---
> **References**
>
> [1] Dick et al., “How many random restarts are enough”. Technical report, 2014.
>
> [2] Jason D. Lee , Max Simchowitz , Michael I. Jordan, and Benjamin Recht, “Gradient Descent Converges to Minimizers”, 9th Annual Conference on Learning Theory, PMLR 49:1246-1257, 2016.
>
> [3] Wachter et al., "Counterfactual explanations without opening the black box: Automated decisions and the GDPR." Harv. JL & Tech., 2017.
>
> [4] Martin Pawelczyk, Chirag Agarwal, Shalmali Joshi, Sohini Upadhyay, and Himabindu Lakkaraju.Exploring counterfactual explanations through the lens of adversarial examples: A theoretical and empirical analysis. In International Conference on Artificial Intelligence and Statistics (AISTATS), 2022.

---

### Official Review · Reviewer_Fd93 · 2022-10-28

**Confidence:** 2
**Clarity, Quality, Novelty And Reproducibility:** See previous answer.
**Correctness:** 4
**Technical Novelty And Significance:** 2
**Empirical Novelty And Significance:** 2
**Recommendation:** 6

**Strength And Weaknesses:**

Strengths:

1) The newly introduced model appears to be a very natural way to combine the two goals, i.e., low cost and robustness.
2) The experiments seem validating enough.

Weaknesses:

1) The clarity in the presentation of the theoretical results is poor. In particular, Section 4.2 is very confusing. Upon reading the earlier sections, I was expecting a theorem that would clearly present the achieved algorithmic trade-off between the confidence parameter r and the cost of the counterfactual.

**Summary Of The Paper:**

The paper considers a model for providing counterfactual explanations that are both easily attainable (i.e., have low cost) and robust (i.e. if the recourse action is not implemented exactly - in other words is noisy - then the output will most likely not change). Due to a natural trade-off between these two criteria, prior work only considers optimizing one at the expense of the other. To my understanding, this paper gives the first result combining both in a controllable way.

The theoretical results of this paper are also accompanied by a validating suite of experiments.

**Summary Of The Review:**

The proposed model and results seem like a interesting contribution to a practical problem. However, the clarity of the writing does not allow me to give a higher score, since I cannot really grasp some of the vital theoretical results of the paper.

---

> ### Author Response · Authors · 2022-11-17
> **Response to Reviewer Fd93**
>
> We thank the reviewer for their insightful feedback and are very grateful that the reviewer appreciates the novelty and significance of our work. Please find our responses below.
>
> > Upon reading the earlier sections, I was expecting a theorem that would clearly present the achieved algorithmic trade-off between the confidence parameter $r$ and the cost of the counterfactual.
>
> Thank you for your feedback. Please note that Proposition 2 provides an expression for the cost of recourse for any given invalidation rate $r$. Intuitively, for any target invalidation rate $r$ we establish a corresponding expression for the cost of recourse. This result is important as it shows that a cost robustness tradeoff exists. In response to your comment, we wrote a new Proposition 3 which builds on the results of Proposition 2. Intuitively Proposition 3 shows that an infinitesimal small increase to target robustness increases recourse costs by a factor that is determined by the level of target robustness and $\sigma$; more formally, a small (i.e., infinitesimal) increase in target robustness (i.e.,$1-r$) increases the cost of recourse (i.e., $\lVert \delta_{E} \rVert_2^2$) by $ \frac{\sigma}{\omega} \frac{1}{\phi(z)}>0$ with $z=\Phi(1-r)^{-1}$ and $r \in (0,1)$ and where $\Phi$ is the Gaussian CDF and $\phi$ denotes the corresponding pdf.

---

### Official Review · Reviewer_fDb4 · 2022-10-31

**Confidence:** 2
**Correctness:** 3
**Technical Novelty And Significance:** 3
**Empirical Novelty And Significance:** 2
**Recommendation:** 6

**Clarity, Quality, Novelty And Reproducibility:**

This paper is very clear. The proposed approach is technically sound overall. However, I have a concern about its ability to explore a significant portion of the cost-invalidation rate Pareto front. Overall, I believe the quality of this work is reasonably high. However, there seems to be little technical novelty. The code to reproduce the experiments was included, but some details of the empirical evaluation seem to be missing. Thus, this paper performs moderately in terms of reproducibility.

**Strength And Weaknesses:**

Strengths:
1. The problem tackled by this problem is practically relevant and, to my knowledge, has not been addressed before.
2. The proposed approach is technically sound overall.
3. This paper is very well-written and easy to follow.
4. The code the reproduce the experiments is available.

Weaknesses:
1. My main concern with the proposed approach is its ability to thoroughly explore the cost-invalidation rate Pareto front. From a multi-objective optimization perspective, this work proposes to use a particular scalarization. However, it is unclear if the entire Pareto front can be recovered by tweaking the parameters of this scalarization. I would like the authors to discuss this.
2. I am not very familiar with the related literature, but the proposed approach seems to have little technical novelty.
3. Is there any guidance to choose $\lambda$ in practice? For example, how is $\lambda$ set in the experiments described in Table 1?
4. Noise in the empirical evaluation seems to be very low ($\sigma^2 = 0.01$). How was this value chosen? The effect of the noise should be investigated more thoroughly.

Other minor comments:
1. The domain of $d_c$ should be $\mathbb{R}^d\times \mathbb{R}^d$ instead of $\mathbb{R}^d$.
2. In the following line, "...is the recourse invalidation rate from equation 1", "equation 1" should be "equation 2".

**Summary Of The Paper:**

This work proposes a probabilistic framework for navigating the trade-off between cost and robustness in algorithmic recourse. More concretely, given a user-specified recourse invalidation rate, a loss function accounting for the invalidation rate, target score, and cost is minimized to find the suggested recourse. One challenge that arises when pursuing this approach is that a naive Monte Carlo approximation of the invalidation rate is non-differentiable. This is circumvented by taking a first-order approximation of the invalidation rate, which has a differentiable closed-form expression. The cost required to achieve a given invalidation rate is derived using this expression. Finally, an upper bound on the (approximate) invalidation rate is derived. The proposed approach is evaluated empirically across various real-world data sets, demonstrating its ability to find better points across the cost-invalidation rate Pareto front than existing approaches.

**Summary Of The Review:**

This work proposes a framework for navigating the trade-off between cost and robustness in algorithmic recourse, a problem that is practically relevant and has not been addressed before. The proposed approach is technically sound. However, I have some concerns about its ability to explore this trade-off fully. Overall this paper is well executed. However, the technical novelty it provides seems limited. I am open to changing my score after the authors' rebuttal and discussion with my fellow reviewers.

---

> ### Author Response · Authors · 2022-11-17
> **Response to reviewer fDb4 (Part 1)**
>
> We thank the reviewer for their insightful comments and suggestions. Please find our responses below.
>
> > [...] it is unclear if the entire Pareto front can be recovered by tweaking the parameters of this scalarization. I would like the authors to discuss this.
>
> The problem of finding recourse does not always benefit from covering the whole Pareto frontier. To provide a concrete example, consider assigning zero weight to the second term from equation (3): such a weighting encourages solutions that have (almost) no cost of recourse and high recourse invalidation rates, while still belonging to the undesirable class. Hence, such solutions are not meaningful for the problem of suggesting actionable algorithmic recourse. Given this, our scalarization focuses on a pareto frontier that encourages meaningful and actionable algorithmic recourse. For example, the second term in the objective from equation (3) encourages changing the predictive outcome from an undesirable prediction (e.g., loan rejection) to a desired prediction (e.g., loan approval). For meaningful recourse we always want this to happen, and thus our current scalarization encourages us to exclude the part of the frontier where the predicted outcome does not change. In summary, we formulated our objective to consider the desired parts of the pareto frontier which allow for low invalidation rates and the flip of the model prediction towards the desirable outcome.
>
> > Is there any guidance to choose $\lambda$ in practice? For example, how is $\lambda$ set in the experiments described in Table 1?
>
> We follow an established procedure to choose the tradeoff parameter $\lambda$. For a given instance and a set of tradeoff parameters (i.e., $\lambda \in$ {0, 0.25, 0.5, 0.75, 1}), we optimize the recourse problem in equation (3) for each tradeoff parameter value. Across all such solutions, we take the solution that results in the least recourse cost. This procedure is consistent with related works [1-5]. In response to your question, we included this detailed description in Section 5 (see updated manuscript).
>
> > Noise in the empirical evaluation seems to be very low ($\sigma^2=0.01$). How was this value chosen? The effect of the noise should be investigated more thoroughly.
>
> Relative to the min-max normalization of the input data (i.e., $x \in [0,1]^d$) used across all our experiments the value $\sigma^2=0.01$ is relatively large, and not small. To better see this consider the standard deviation $\sigma=0.1$, which covers 10% percent of the input range. For example, for $\sigma^2=0.01$ PROBE suggests robust recourses with respect to noisy responses that have a $10$ percent margin of error. Also note that related work chose noise magnitudes of $0.1$ while working with standard-scaled data (i.e., $x \in [-3.5, 3.5]$) [4]. Such a noise magnitude effectively covers 1.5 percent of the input range, and is much smaller than in our work.
>
> Appendix B demonstrates results for an even larger $\sigma^2=0.025$. For this larger noise magnitude PROBE performs as expected: the main difference compared to choosing $\sigma^2=0.01$ is that the recourse costs are now larger (see Table 5 in Appendix C), which is in line with the result of Proposition 2 and thus as expected.
>
> ---
> **References**
>
> [1] Sandra Wachter, Brent Mittelstadt, and Chris Russell. Counterfactual explanations without opening the black box: automated decisions and the gdpr. Harvard Journal of Law & Technology, 31(2), 2018.
>
> [2] Ramaravind K. Mothilal, Amit Sharma, and Chenhao Tan. Explaining machine learning classifiers through diverse counterfactual explanations. In Proceedings of the Conference on Fairness, Accountability, and Transparency (FAT*), 2020.
>
> [3] Martin Pawelczyk, Sascha Bielawski, Johan Van den Heuvel, Tobias Richter, and Gjergji Kasneci. Carla: A python library to benchmark algorithmic recourse and counterfactual explanation algorithms. In Advances in Neural Information Processing Systems (NeurIPS) (Benchmark and Datasets Track), volume 34, 2021.
>
> [4] Sohini Upadhyay, Shalmali Joshi, and Himabindu Lakkaraju. Towards robust and reliable algorithmic recourse. In Advances in Neural Information Processing Systems (NeurIPS), volume 34, 2021.
>
> [5] Ricardo Dominguez-Olmedo, Amir-Hossein Karimi, and Bernhard Schölkopf. On the adversarial ¨ robustness of causal algorithmic recourse. In International Conference on Machine Learning (ICML), 2022.

---

> > ### Author Response · Authors · 2022-11-17
> > **Response to reviewer fDb4 (Part 2)**
> >
> > [Continued from Part 1 above]
> >
> > > I am not very familiar with the related literature, but the proposed approach seems to have little technical novelty.
> >
> > Prior works by Upadhyay et al [1] and Dominguez-Olmedo et al [2] attribute robustness to recourses in different ways. While the former constructs recourses that are robust to small shifts in the underlying model, the latter constructs recourses that are robust to small input perturbations. These approaches adapt the classic minimax objective functions commonly employed in adversarial robustness and robust optimization literature to the setting of algorithmic recourse, and use gradient descent style approaches to optimize these functions. In an attempt to generate recourses that are robust to either small shifts in the model (Upadhyay et. al., [1]) or to small input perturbations (Dominguez-Olmedo et. al. [2]), the above approaches find recourses that are farther away from the underlying model's decision boundaries, thereby increasing the distance between the counterfactuals (recourses) and the original instances (i.e., the recourse costs). Higher cost recourses are harder to implement for end users as they are farther away from the original instance vectors (current user profiles). Putting it all together, the aforementioned approaches generate robust recourses that are often high in cost and are therefore harder to implement, without providing end users with any say in the matter.
> >
> > In contrast, our work puts forth a paradigm shifting idea of enabling users to control the recourse robustness-cost tradeoffs by letting them choose the probability with which a recourse could get invalidated (recourse invalidation rate) if small changes are made to the recourse i.e., the recourse is implemented somewhat noisily. Given this problem formulation, we can no longer use the minimax objectives outlined by prior works as we need to ensure that the resulting recourse invalidation rates match desired invalidation rates input by end users. To this end, we propose a novel objective function (Eqn. 3 in Section 4 of the main paper) which simultaneously minimizes the gap between the achieved (resulting) and desired recourse invalidation rates, minimizes recourse costs, and also ensures that the counterfactual (recourse) achieves a positive model prediction.
> >
> > To optimize the proposed objective, we outline a gradient descent style approach (Algorithm 1, main paper). Note that we need to compute the achieved recourse invalidation rate at each step of the gradient descent algorithm, and computing this empirically can be computationally prohibitive as demonstrated by prior work (e.g., Slack et. al., [3]) since it involves generating perturbations of each candidate counterfactual and querying the underlying model for labels of all the perturbations. To this end, we develop novel theoretical expressions for the recourse invalidation rates (Theorems 1, 2 in main paper and Appendix) corresponding to any given instance w.r.t. different classes of underlying models (e.g., linear models, non-linear models such as deep neural networks and tree based models), and then use these estimates to efficiently optimize the proposed objective.
> >
> > Note that the approaches put forth by prior works only handle linear models or local linear model approximations. In contrast, the approaches and theoretical derivation of the invalidation rate we propose (Section 4 and Appendix A) enable us to handle both linear and non-linear models (e.g., deep neural networks, tree based models) effectively. Our empirical results (Figure 4) also demonstrate that our approach achieves better recourse cost/invalidation rate tradeoffs compared to both Upadhyay et al [1] and Dominguez-Olmedo et al [2].
> >
> > In **summary**, the following are the **key novel contributions** of our work: (i) proposing a novel problem formulation which enables end users to manage the recourse robustness-cost tradeoffs, (ii) introducing new theoretical results to bound the recourse invalidation rates (Theorems 1, 2) corresponding to any given instance w.r.t. various classes of underlying models (e.g., linear models, non-linear models such as deep nets, tree) and (iii) developing a novel objective function (Eqn. 3 in Section 4 of the main paper) which leverages the above theoretical estimates to ensure that the resulting invalidation rates match the user preferred invalidation rates.
> >
> > ---
> > **References**
> >
> > [1] Sohini Upadhyay, Shalmali Joshi, and Himabindu Lakkaraju. Towards robust and reliable algorithmic recourse. In NeurIPS, 2021.
> >
> > [2] Ricardo Dominguez-Olmedo, Amir-Hossein Karimi, and Bernhard Schölkopf. On the adversarial robustness of causal algorithmic recourse. In ICML, 2022.
> >
> > [3] Dylan Slack, Sophie Hilgard, Himabindu Lakkaraju, and Sameer Singh. Counterfactual explanations can be manipulated. In NeurIPS, 2021

---

> > > ### Comment · Reviewer_fDb4 · 2022-11-23
> > > **Reviewer fDb4 post-rebuttal update**
> > >
> > > Dear authors,
> > >
> > > Thank you for your detailed response. All my concerns have been addressed, except for one: the ability of the proposed approach to explore the Pareto front. I understand why exploring the entire front may be unnecessary (and perhaps detrimental). However, my concern is that, depending on the shape of the Pareto front, the proposed approach may recover only a tiny portion of it. (In multi-objective optimization, using linear scalarizations to explore a concave Pareto front recovers only the extremum points.) In such scenarios, your approach may not be exploring the cost-invalidation rate trade-off at all, which seems to me like a big drawback. I kindly ask you to do your best to say something about this in the final version of your work.
> > >
> > > Given that you have successfully addressed most of my concerns, and other reviewers (probably more knowledgeable than me on this topic) show optimism about your work, I have decided to raise my score slightly.
> > >
> > > Best wishes,
> > > Reviewer fDb4

---

### Decision · Program_Chairs · 2023-01-20

**Decision:**

Accept: poster

**Justification For Why Not Higher Score:**

The paper does not rise to the level of oral/spotlight: e.g., the authors do not consider the whole Pareto frontier of the tradeoff.

**Justification For Why Not Lower Score:**

Impressive combination of theory and experimentation for an increasingly-important problem domain in explainable AI.

**Metareview: Summary, Strengths And Weaknesses:**

It has become clear that in AI/ML—driven decisions (e.g., in loan decisions), individuals who are adversely impacted (e.g., their loan application getting denied by the ML system) are provided a means for recourse. Known approaches achieve low cost---ease of implementation---or robustness to small perturbations, but not both due to the inherent trade-offs between the cost and robustness. Also, these known approaches do not provide users with the ability to control these trade-offs. This paper develops what may be the first rigorous algorithmic framework for users to manage this trade-off. Specifically, the paper’s Probabilistically ROBust rEcourse (PROBE) approach lets end-users choose the probability with which a recourse could get invalidated if the recourse is implemented somewhat noisily. Both rigorous theory and promising experimental results are presented.

The paper is very well-written. The authors are encouraged to show better how the proposed approach can explore the whole Pareto frontier for the tradeoff.


**Note From Pc:**

if the above contains the word "oral" or "spotlight" please see: "oral" presentation means -> notable-top-5% and "spotlight" means -> notable-top-25%. As stated in our emails, we are disassociating presentation type from AC recommendations